# MemSim: A Bayesian Simulator for Evaluating Memory of LLM-based Personal Assistants

**Zeyu Zhang**[1][†][*]♠♣**, Quanyu Dai**[2][*]**, Luyu Chen**[1]♠♣**, Zeren Jiang**[3]**, Rui Li**[1]♠♣**, Jieming Zhu**[2]**,**
**Xu Chen**[1][§]♠♣**, Yi Xie**[3]**, Zhenhua Dong**[2]**, Ji-Rong Wen**[1]♠♣

[1]Gaoling School of Artificial Intelligence, Renmin University of China
[2]Huawei Noah's Ark Lab    [3]Huawei Technologies Ltd.
{zeyuzhang,xu.chen}@ruc.edu.cn, daiquanyu@huawei.com

## Abstract

LLM-based agents have been widely applied as personal assistants, capable of memorizing information from user messages and responding to personal queries. However, there still lacks an objective and automatic evaluation on their memory capability, largely due to the challenges in constructing reliable questions and answers (QAs) according to user messages. In this paper, we propose MemSim, a Bayesian simulator designed to automatically construct reliable QAs from generated user messages, simultaneously keeping their diversity and scalability. Specifically, we introduce the Bayesian Relation Network (BRNet) and a causal generation mechanism to mitigate the impact of LLM hallucinations on factual information, facilitating the automatic creation of an evaluation dataset. Based on MemSim, we generate a dataset in the daily-life scenario, named MemDaily, and conduct extensive experiments to assess the effectiveness of our approach. We also provide a benchmark for evaluating different memory mechanisms in LLM-based agents with the MemDaily dataset. To benefit the research community, we have released our project at `https://github.com/nuster1128/MemSim`.

## 1 Introduction

In recent years, large language model (LLM) based agents have been extensively deployed across various fields [1–6]. One of their most significant applications is serving as personal assistants [7, 8], where they engage in long-term interactions with users to address a wide range of issues [9, 10]. For LLM-based personal assistants, memory is one of the most significant capability [11]. To perform personal tasks effectively, these agents must be capable of storing factual information from previous messages and recalling relevant details to generate appropriate responses. For example, a user Alice might tell the agent, "*I will watch a movie at City Cinema this Friday in Hall 3, Row 2, Seat 9.*" When Friday arrives, she might ask the agent, "*Where is my movie seat?*" Then, the agent should recall the relevant information (i.e., the seat number) to generate an appropriate response to Alice.

Previous research has proposed methods for constructing the memory of LLM-based agents [12, 13, 9, 14, 15]. However, there remains a lack of objective and automatic methods to evaluate how well personal assistants can memorize and utilize factual information from previous messages. One conventional solution is to collect messages from real-world users, and manually annotate answers to human-designed questions based on these messages. However, it requires substantial human labor

---

[†] Work under the internship in Huawei Noah's Ark Lab.
[*] Co-first authors.
[§] Corresponding author.
♠ Beijing Key Laboratory of Research on Large Models and Intelligent Governance.
♣ Engineering Research Center of Next-Generation Intelligent Search and Recommendation, MOE.

39th Conference on Neural Information Processing Systems (NeurIPS 2025).

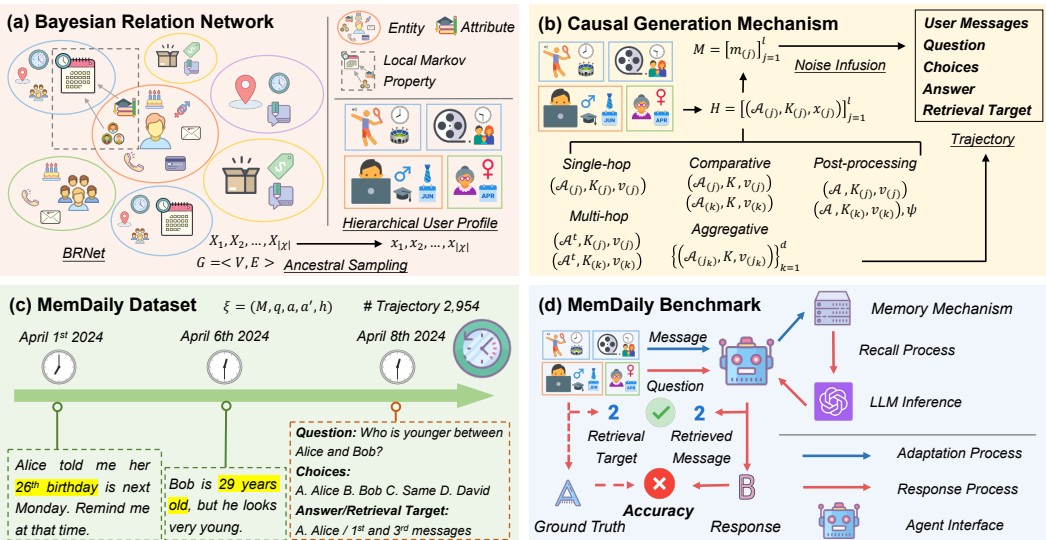

Figure 1: The overview of MemSim framework and MemDaily dataset.

that lacks **scalability**. Another solution is to generate user messages and question-answers (QAs) with LLMs. However, the hallucination of LLMs can severely undermine the **reliability** of generated datasets, particularly in complex scenarios [16]. Here, we refer to the reliability of a dataset as the correctness of its ground truths to factual questions given the corresponding user messages. Our research shows that due to the hallucination of LLMs, the correctness of ground truths generated by vanilla LLMs is less than 90% in most scenarios and can fall below 40% in some complex scenarios (see **Section 5.2**). Moreover, generating diverse user profiles through LLMs is also challenging, as they tend to produce the most plausible profiles that lack **diversity**.

To address these challenges, we propose MemSim, a Bayesian simulator designed to construct reliable QAs from generated user messages, keeping their diversity and scalability. It can be utilized to evaluate the memory capability of LLM-based personal assistants. Specifically, we introduce the Bayesian Relation Network (BRNet) to generate the simulated users that are represented by their hierarchical profiles. Then, we propose a causal generation mechanism to produce various types of user messages and QAs for the comprehensive evaluation on memory mechanisms. Based on MemSim, we create a dataset in the daily-life scenario, named MemDaily, and perform extensive experiments to assess the quality of MemDaily. Finally, we construct a benchmark to evaluate different memory mechanisms of LLM-based agents with MemDaily. Our work is the first one that evaluates memory of LLM-based personal assistants in an objective and automatic way. Our contributions are summarized as follows:

• We analyze the challenges of constructing datasets for objective evaluation on the memory capability of LLM-based personal assistants, focusing on the aspects of reliability, diversity, and scalability.

• We propose MemSim, a Bayesian simulator designed to generate reliable, diverse and scalable datasets for evaluating the memory of LLM-based personal assistants. We design BRNet to generate the simulated users, and propose a causal generation mechanism to construct user messages and QAs.

• We create the MemDaily dataset, which can be used to evaluate the memory capability of LLM-based personal assistants. We perform extensive experiments to assess the quality of MemDaily, and provide a benchmark for different memory mechanisms of LLM-based agents. To support the research community, we release project available at `https://github.com/nuster1128/MemSim`.

The rest of our paper is organized as follows. In **Section 2**, we review the related works on the evaluation of memory in LLM-based agents and personal assistants. In **Section 3**, we introduce the details of MemSim, and the generation process of MemDaily. In **Section 4**, we assess the quality of MemDaily. **Section 5** provides a benchmark for evaluating different memory mechanisms of LLM-based agents. Finally, in **Section 6**, we discuss the limitations of our work and draw conclusions.

## 2   Related Works

LLM-based agents have been extensively utilized across various domains, marking a new era for artificial personal assistants [7, 17, 18]. For LLM-based personal assistants, memory is critical to

enable agents to deliver personalized services. This includes storing, managing, and utilizing users' personal and historical data [11, 12, 15, 19–23]. For instance, MPC [10] suggests storing essential factual information in a memory pool with a summarizer for retrieval as needed. MemoryBank [12] converts daily events into high-level summaries and organizes them into a hierarchical memory structure for future retrieval. These approaches primarily aim to enhance agents' memory capability.

Previous studies have also attempted to evaluate the memory capability of LLM-based agents. Some studies use subjective methods, employing human evaluators to score the effectiveness of retrieved memory [10, 12, 24]. Other studies use objective evaluations by constructing dialogues and question-answer pairs [14, 25, 26]. However, there exist limitations in higher cost and lower reliability. Some previous studies construct knowledge-based question-answering datasets to assess Retrieval-Augmented Generation [27, 28]. These studies either use knowledge graphs to generate QAs through templates or manually annotate QAs with human input [29–34]. However, most of them focus on common-sense questions rather than personal questions, and they do not include textual user messages and target indexes for retrieval evaluation [30–32, 35, 36]. They are also highly dependent on the entities extracted from the given corpus [30, 35]. Our work is the first that evaluate memory of LLM-based personal assistants in an objective and automatic way, which can generate user messages and QAs without human annotators, keeping reliability, diversity and scalability.

## 3 Methods

### 3.1 Overview of MemSim

In order to construct reliable QAs from generated user messages, we propose a Bayesian simulator named MemSim shown in **Figure 1**. First, we develop the Bayesian Relation Network to model the probability distribution of users' relevant entities and attributes, enabling the sampling of diverse hierarchical user profiles. Then, we introduce a causal mechanism to generate user messages and construct reliable QAs. We design various types of QAs for comprehensive memory evaluation, incorporating different noises to simulate real-world environments. Based on the constructed QAs and generated user messages, researchers can objectively and automatically evaluate the memory capability of LLM-based personal assistants on factual information from previous messages.

### 3.2 Bayesian Relation Network

We introduce Bayesian Relation Network (BRNet) to model the probability distribution of users' relevant entities and attributes, where we sample hierarchical profiles to represent simulated users (see **Figure 1(a)**). Specifically, we define a two-level structure in BRNet, including the entity level and the attribute level. The entity level represents user-related entities, such as relevant persons, involved events, and the user itself. At the attribute level, each entity comprises several attributes, such as age, gender, and occupation. Here, BRNet actually serves as a predefined meta-user. Formally, let $\mathcal{A}^1, \ldots, \mathcal{A}^N$ represent different entities, and each entity $\mathcal{A}^i$ comprises several attributes $\{A_1^i, A_2^i, \ldots, A_{N^i}^i\}$, where $N$ is the number of entities, and $N^i$ is the number of attributes belonging to the entity $\mathcal{A}^i$. Each attribute $A_j^i$ corresponds to a random variable $X_j^i$, which can be sampled in a value space. For example, the *college's* (entity $\mathcal{A}^i$) *age* (attribute $A_j^i$) is *28 years old* (value $x_j^i \sim X_j^i$).

We denote BRNet as a directed graph $G = \langle V, E \rangle$ at the attribute level, where the vertex set $V$ includes all attributes, i.e., $V = \bigcup_{i=1}^{N}\{A_1^i, A_2^i, \ldots, A_{N^i}^i\}$. The edge set $E$ captures all the direct causal relations among these attributes, defined as $E = \{\langle A_j^i, A_l^k \rangle \mid \forall X_j^i, X_l^k \in \mathcal{X}, X_j^i \to X_l^k\}$, where $\mathcal{X} = \bigcup_{i=1}^{N}\{X_1^i, X_2^i, \ldots, X_{N^i}^i\}$. For better demonstration, in this subsection, we simplify the subscripts of the variables in $\mathcal{X}$ as $1, 2, ..., \sum_{i=1}^{N} N_i$. The conditional probability distribution among them can either be explicitly predefined or implicitly represented by LLM's generation with conditional prompts. It is important to note that we assume the causal structure is loop-free, ensuring that BRNet forms a directed acyclic graph (DAG), which is typical in most scenarios [37]. Additionally, the vertices (i.e., attributes), edges (i.e., causal relations), and conditional probability distributions (i.e., prior knowledge) can be easily scaled to different scenarios.

So far, we have constructed the BRNet, where the joint probability distribution $P(X_1, X_2, \ldots, X_{|\mathcal{X}|})$ over all attributes can represent the user distribution in the given scenario. Then, we can sample different values of attributes on entities from BRNet to represent various user profiles. One straightforward approach is to compute the joint probability distribution and sample from it.

**Assumption 1** (Local Markov Property). *BRNet satisfies the local Markov property, which states that*

$$X_t \perp\!\!\!\perp X_{\overline{des}(X_t)} | par(X_t), \forall X_t \in \mathcal{X},$$

*where $\overline{des}(X_t)$ denotes the non-descendant set of $X_t$, $par(X_t)$ denotes the parent set of $X_t$, and the notation $\cdot \perp\!\!\!\perp \cdot | \cdot$ indicates the variables are conditionally independent.*

Because the parents of an attribute can be extended to any non-descendant attributes of it by adding a new edge if they have a direct causal relation. Given these parent attributes, other non-descendent attributes are conditionally independent of that attribute according to Pearl [38].

**Theorem 1** (Factorization). *The joint probability distribution of BRNet can be expressed as*

$$P(X_1, X_2, ..., X_{|\mathcal{X}|}) = \prod_{X_t \in \mathcal{X}} P(X_t | par(X_t)),$$

*where $par(X_t)$ denotes the set of parent attributes of $X_t$.*

The proof of this theorem is provided in **Appendix A.1**. However, calculating the joint probability distribution and sampling from it may be impractical in our scenarios. First, the joint probability distribution is often high-dimensional, making its calculation and sampling costly. Second, some conditional probability distributions are difficult to represent in explicit forms, particularly when using LLMs for value generation through conditional prompts. To address these issues, we introduce the ancestral sampling process to obtain the values of attributes.

**Assumption 2** (Conditional Sampling). *In BRNet, an attribute can be sampled from the conditional probability distribution given its parent attributes. Specifically, we have*

$$\tilde{x}_t \sim P\left(X_t | par\left(X_t\right)\right), \forall X_t \in \mathcal{X},$$

*where the conditional probability distribution can be expressed in either explicit or implicit forms.*

The ancestral sampling algorithm is outlined as follows. First, we obtain the topological ordering of BRNet using Kahn's algorithm [39]. Next, we sample all attributes according to this ordering. For top-level attributes without parents, the sampling is performed based on their marginal probability distributions. For other variables like $X_t$, we sample their values using the conditional probability distribution $\tilde{x}_t \sim P\left(X_t | par\left(X_t\right)\right)$ as specified in **Assumption 2**. Finally, we consider each sampling result $\{\tilde{x}_1, \tilde{x}_2, \ldots, \tilde{x}_{|\mathcal{X}|}\}$ as the attribute-level profiles of a user, which constitute different entities as the entity-level profiles of the user. These two levels represent the user in different grains, which are important to generate user messages and QAs subsequently.

**Theorem 2** (Ancestral Sampling). *For BRNet, the result of ancestral sampling is equivalent to that of sampling from the joint probability distribution. Specifically, we have*

$$P(\tilde{x}_1, \tilde{x}_2, ..., \tilde{x}_{|\mathcal{X}|}) = P(x_1, x_2, ..., x_{|\mathcal{X}|}),$$

*where $x_1, x_2, ..., x_{|\mathcal{X}|} \sim P(X_1, X_2, ..., X_{|\mathcal{X}|})$ are sampled from the joint probability distribution.*

The proof can be found in **Appendix A.2**. By employing ancestral sampling [38], we eliminate the need to compute the joint probability distribution, making the sampling process more efficient and practical. By utilizing BRNet, we introduce prior knowledge of the specific scenario into the graphical structure and sampling process, which can improve the diversity and scalability of user profiles, thereby enhancing the diversity and scalability of whole datasets.

### 3.3 Causal Generation Mechanism

Based on hierarchical user profiles, we propose a causal generation mechanism to generate user messages, and construct reliable QAs corresponding to them. Here, *causal* indicates that the generation of user messages and the construction of QAs are causally dependent on the same informative *hints* that are also causally derived from hierarchical user profiles. Specifically, we define a piece of hint as a triple $(\mathcal{A}^i, A^i_j, x^i_j)$ that provides factual information in a structural format. In other words, the hierarchical user profiles provide a structural foundation to get different hints, which then provide a set of relevant information as the causation of both user messages and QAs, shown in **Figure 1(b)**.

**Construction of Informative Hints.** We construct the hints of factual information based on hierarchical user profiles before creating the user messages and QAs. We select a target entity $\mathcal{A}^t$ at the

Table 1: Overview of comprehensive questions and answers.

| Types | Descriptions | Examples | Causal Hints | Retrieval Target |
|---|---|---|---|---|
| Single-hop | Rely on one message to answer the question directly. | Q: When is Alice's birthday ? A: June 1st. | $(\mathcal{A}_{(j)}, K_{(j)}, v_{(j)})$ | $\{m_{(j)}\}$ |
| Multi-hop | Require multiple messages to answer the question jointly. | Q: Where is the meeting that I will attend next week? A: Victoria Conference Center. | $(\mathcal{A}^t, K_{(j)}, x_{(j)})$, $(\mathcal{A}^t, K_{(k)}, x_{(k)})$ | $\{m_{(j)}, m_{(k)}\}$ |
| Comparative | Compare two entities on a shared attribute with multiple messages. | Q: Who is younger between Alice and Bob? A: Bob. | $(\mathcal{A}_{(j)}, K, v_{(j)})$, $(\mathcal{A}_{(k)}, K, v_{(k)})$ | $\{m_{(j)}, m_{(k)}\}$ |
| Aggregative | Aggregate messages about more than two entities on a common attribute. | Q: How many people are under 35 years old? A: Three. | $\{(\mathcal{A}_{(j_k)}, K, v_{(j_k)})\}_{k=1}^d$ | $\{m_{(j_k)}\}_{k=1}^d$ |
| Post-processing | Involve extra reasoning steps to answer with multiple messages. | Q: What season was the teacher that I know born in? A: Spring. | $(\mathcal{A}^t, K_{(j)}, v_{(j)})$, $(\mathcal{A}^t, K_{(k)}, v_{(k)})$ | $\{m_{(j)}, m_{(k)}\}$ |

Table 2: Summary of the MemDaily dataset.

| Statistics | Simp. | Cond. | Comp. | Aggr. | Post. | Noisy | Total |
|---|---|---|---|---|---|---|---|
| Trajectories | 500 | 500 | 492 | 462 | 500 | 500 | 2,954 |
| Messages | 4215 | 4195 | 3144 | 5536 | 4438 | 4475 | 26,003 |
| Questions | 500 | 500 | 492 | 462 | 500 | 500 | 2,954 |
| TPM | 15.48 | 15.49 | 14.66 | 14.65 | 17.07 | 16.14 | 15.59 |

entity-level, and choose $l^t$ attributes $\{K_1^t, K_2^t, \ldots, K_{l^t}^t\} \subseteq \mathcal{A}^t$ along with their corresponding values $\{v_1^t, v_2^t, \ldots, v_{l^t}^t\}$ from the attribute-level profiles. Then, we reformulate them into a list of triple hints $H^t = [(\mathcal{A}^t, K_i^t, v_i^t)]_{i=1}^{l^t}$. For some complex types of QAs, we choose more than one target entities, and concatenate their lists of hints. For better demonstration, we re-index the final list of hints as $H = \left[(\mathcal{A}_{(j)}, K_{(j)}, v_{(j)})\right]_{j=1}^{l}$, where $l$ is the number of hints in the final list.

**Construction of User Messages.** Based on the $j$-th hint $(\mathcal{A}_{(j)}, K_{(j)}, v_{(j)}) \in H$, we construct the corresponding user message $m_{(j)}$ with LLM, where we have $m_{(j)} = LLM(\mathcal{A}_{(j)}, K_{(j)}, v_{(j)})$. Here, the LLM only serves the purpose of rewriting structural hints, without any reasoning process. For example, if the hint is (*my uncle Bob, occupation, driver*), the generated user message might be "*The occupation of my uncle Bob is a driver*". We generate user messages for all the hints in $H$, and we finally get the list of user messages $M = \left[m_{(j)}\right]_{j=1}^{l}$.

**Construction of Questions and Answers.** In order to evaluate the memory capability of LLM-based personal assistants more comprehensively, we propose to construct five representative types of QAs to cover various complexities in real-world scenarios, as detailed in **Table 1**. For each question $q$, we provide three forms of ground truths: (1) the textual answer $a$ that can correctly respond to $q$, (2) the correct choice $a$ among confusing choices $a'$ (generated by LLM) as a single-choice format, and (3) the correct retrieval target $h \subseteq M$ that contains the required factual information to the question.

*(i.) Single-hop QA.* Single-hop QA is the most basic type of QAs, relying on a single piece message to directly answer the question. In constructing QA, we randomly select the $j$-th hint $(\mathcal{A}_{(j)}, K_{(j)}, v_{(j)})$ and generate the question $q = LLM(\mathcal{A}_{(j)}, K_{(j)})$ through LLM rewriting, where the answer is $a = v_{(j)}$. Correspondingly, the retrieval target is $h = \{m_{(j)}\}$.

*(ii.) Multi-hop QA.* Multi-hop QA necessitates the use of multiple messages to determine the correct answer, making it more complex than single-hop QA. In constructing Multi-hop QA, we first sample two hints $(\mathcal{A}_{(j)}, K_{(j)}, v_{(j)})$ and $(\mathcal{A}_{(k)}, K_{(k)}, v_{(k)})$ from the same bridge entity $\mathcal{A}^t$ (i.e., $\mathcal{A}^t = \mathcal{A}_{(j)} = \mathcal{A}_{(k)}$). We then mask this bridge entity and generate the question $q = LLM(K_{(j)}, v_{(j)}, K_{(k)})$ through LLM rewriting, where the answer is $a = v_{(k)}$. The target message set is $h = \{m_{(j)}, m_{(k)}\}$. By incorporating additional entities, the questions can be easily extended to more hops.

*(iii.) Comparative QA.* Comparative QA is an extensive type of multi-hop QA, which involves comparing two entities based on a shared attribute. We first select two hints $(\mathcal{A}_{(j)}, K_{(j)}, v_{(j)})$ and $(\mathcal{A}_{(k)}, K_{(k)}, v_{(k)})$ from different entities with the same meaning attribute $K$ (i.e., $\mathcal{A}_j \neq \mathcal{A}_k$ and $K \cong$

$K_{(j)} \cong K_{(k)}$). We then rewrite the question $q = LLM(\mathcal{A}_{(j)}, \mathcal{A}_{(k)}, K)$ by LLM, where the answer $a = f(K, v_{(j)}, v_{(k)})$ is derived from the function $f(\cdot)$. The retrieval target is $h = \{m_{(j)}, m_{(k)}\}$.

*(iv.) Aggregative QA.* Aggregative QA is a general type of comparative QA, which requires aggregating messages from more than two entities on a shared attribute. For construction, we choose $d$ hints $\{(\mathcal{A}_{(j_k)}, K, v_{(j_k)})\}_{k=1}^d$ from different entities with the same meaning attribute $K$. Then, we construct the question $q = LLM(\{\mathcal{A}_{(j_k)}\}_{j=1}^d, K)$, where we obtain the answer $a = f(K, \{v_{(j_k)}\}_{k=1}^d)$. The target message set should include all these related references, that is, $h = \{m_{(j_k)}\}_{k=1}^d$.

*(v.) Post-processing QA.* Post-processing QA addresses situations where personal questions require additional reasoning steps for agents to answer, based on the retrieved messages. We first select two hints $(\mathcal{A}_{(j)}, K_{(j)}, v_{(j)})$ and $(\mathcal{A}_{(k)}, K_{(k)}, v_{(k)})$ from the same bridge entity $\mathcal{A}^t$. We then design a reasoning factor $\psi$ to generate the question $q = LLM(K_{(j)}, v_{(j)}, K_{(k)}, \psi)$, and derive the answer $a = f(K_{(k)}, v_{(k)}, \psi)$, where $\psi$ specifies the reasoning process. For example, it could be "*the sum of the last five digits of the phone number $v_{(k)}$*". Similarly, the retrieval target will be $h = \{m_{(j)}, m_{(k)}\}$.

**Infusion of Noise in User Messages.** We integrate two types of noise in user messages by concatenation, in order to simulate real-world circumstances. The first type is entity-side noise, which refers to noisy messages that contain the selected attributes from unselected entities. The second type is attribute-side noise, which involves noisy messages that describe unselected attributes of the selected entities. Both types of noise can impact agents' ability to retrieve messages and generate answers.

Eventually, we formulate the trajectory $\xi = (M, q, a, a', h)$ by discarding all hints, where each trajectory serves as a test instance for evaluating the memory capability of LLM-based personal assistants. There are two insights into the causal generation mechanism. First, the factual information of messages and QAs are causally constructed from the shared hints that are sampled from user profiles, where LLMs are only responsible for rewriting based on the given information, rather than imagining or reasoning. This pipeline mitigates the impact of LLM hallucination on the factual information, keeping the reliability of QAs. It can also prevent contradictions among user messages from the same trajectory, because their hints are derived from the same user profile. Second, our method focuses on designing the asymmetric difficulty between constructing QAs (i.e., profiles→hints→messages, question and answer) and solving QAs (i.e., messages|question→answer), which is critical for the automatic generation of evaluation datasets.

### 3.4 MemDaily: A Dataset in the Daily-life Scenario

Based on MemSim, we create a dataset in the daily-life scenario, named MemDaily, which can be used to evaluate the memory capability of LLM-based personal assistants, shown in **Figure 1(c)**. Specifically, MemDaily incorporates 11 entities and 73 attributes (see details in **Appendix G.1**), all of which are representative and closely related to users' daily lives. We create 6 sub-datasets of different QA types mentioned previously: (1) **Simple (Simp.)**: single-hop QAs. (2) **Conditional (Cond.)**: multi-hop QAs with conditions. (3) **Comparative (Comp.)**: comparative QAs. (4) **Aggregative (Aggr.)**: aggregative QAs. (5) **Post-processing (Post.)**: post-processing QAs. (6) **Noisy**: multi-hop QAs with additional irrelevant noisy texts inside questions. The summary of MemDaily is shown in **Table 2**, where we present the number of trajectories, user messages, questions, and TPM (tokens per message). More details and examples can be found in **Appendix G**.

## 4 Experiments of Evaluation

In this section, we evaluate the quality of MemDaily, which can reflect the effectiveness of MemSim. Specifically, the evaluations are conducted in three parts: the user profiles, the user messages, and the constructed QAs. Besides, we also conduct comprehensive case studies in **Appendix G**.

### 4.1 Evaluation on User Profiles

The generated user profiles are supposed to express both rationality and diversity, which also directly influence the creation of user messages and QAs. Therefore, we evaluate these two aspects to reflect their quality. Rationality means that the user profiles should possibly exist in the real world, with no internal contradictions in their descriptions. Diversity indicates that the descriptions among users are distinct, covering a wide range of user types.

Table 3: Results of the evaluation on user profiles.

| Methods | R-Human | R-GPT | SWI-R | SWI-O | SWI-A |
|---------|---------|-------|-------|-------|-------|
| IndePL | 1.35±0.53 | 4.32 | 0.464 | 0.231 | 0.347 |
| SeqPL | 1.64±0.73 | 4.40 | 1.471 | 1.416 | 1.443 |
| JointPL | 3.02±1.14 | **4.80** | 1.425 | 0.462 | 0.943 |
| MemSim | **4.91±0.30** | 4.68 | **3.206** | **2.895** | **3.050** |

**Metrics.** For rationality, we recruit six human evaluators to score the generated user profiles on a scale from 1 to 5 based on the guidelines in **Appendix H.1**. Additionally, we use GPT-4o [1] as a reference for scoring. These two metrics are denoted as R-Human and R-GPT. For diversity, we calculate the average Shannon-Wiener Index (SWI) [40] on key attributes, using the following formula:

$$\text{SWI-}\mathcal{W} = -\frac{1}{|\mathcal{W}|} \sum_{X_k \in \mathcal{W}} \sum_{x_i \in X_k} p(x_i) \ln p(x_i),$$

where $\mathcal{W} \subseteq \mathcal{X}$ is the subset of attribute variables. Therefore, we calculate SWI-R, SWI-O, and SWI-A, corresponding to role-relevant attributes, role-irrelevant attributes, and all attributes, respectively.

**Baselines.** We design several baselines to generate user profiles: (1) **JointPL**: prompting an LLM to generate attributes jointly. (2) **SeqPL**: prompting an LLM to generate attributes sequentially, conditioned on previous attributes in linear order. (3) **IndePL**: prompting an LLM to generate attributes independently. We compare our method with these baselines on generating user profiles.

**Results.** As shown in **Table 3**, MemSim outperforms other baselines on R-Human, demonstrating the effectiveness of BRNet as an ablation study. However, we also observe an inconsistency between R-Human and R-GPT, which may be due to the inaccuracy of the LLM's scoring [41]. Furthermore, our method achieves the highest diversity compared to the other baselines.

## 4.2 Evaluation on User Messages

We evaluate the quality of generated user messages in multiple aspects, including fluency, rationality, naturalness, informativeness, and diversity. The first four aspects are designed to assess the quality inside a trajectory, while the final one targets the variety across trajectories.

**Metrics.** For the inside-trajectory aspects, human evaluators score user messages on a scale from 1 to 5 based on the guidelines in **Appendix H.2**, denoted as **F-Human** (fluency), **R-Human** (rationality), **N-Human** (naturalness), and **I-Human** (informativeness). GPT-4o scores are also available and detailed in **Appendix D**. To assess the diversity across trajectories, we extract all entities and calculate their average Shannon-Wiener Index per 10,000 tokens of user messages, referred to as **SWIP**.

**Baselines.** We implement several baselines that generate messages under different constraints regarding user profiles and tasks: (1) **ZeroCons**: no constraints on attributes when prompting LLMs. (2) **PartCons**: partial attributes of user profiles are constrained in prompts for LLMs. (3) **SoftCons**: full attributes of user profiles are constrained in prompts but they are not forcibly for generation. Our MemSim method imposes the most strict constraints, requiring both the integration of specific attributes into user messages and ensuring that questions are answerable with established ground truths based on the shared hints. Generally, higher constraint commonly means sacrifice of fluency and naturalness, because it compulsively imposes certain information to benefit QA constructions.

**Results.** As shown in **Table 4**, our method maintains relatively high scores despite the rigorous constraints on constructing reliable QAs. Additionally, MemSim exhibits the highest diversity index, attributed to the BRNet and the causal generation mechanism that produces a wider variety of user messages based on the provided hierarchical user profiles.

## 4.3 Evaluation on Questions and Answers

The primary challenge for constructing a reliable dataset is ensuring the accuracy of ground truths for the constructed questions. To assess the reliability of MemDaily, we sample approximately 20% of all the trajectories in MemDaily and employ human evaluators to verify the correctness of their ground truths. Specifically, the evaluators are required to examine three parts of the ground truths: textual

---

[1] https://openai.com/index/hello-gpt-4o/

Table 4: Results of the evaluation on user messages.

| Methods | F-Human | R-Human | N-Human | I-Human | SWIP |
|---------|---------|---------|---------|---------|------|
| ZeroCons | 4.94±0.24 | **4.94±0.24** | 4.85±0.35 | 2.82±1.15 | 2.712 |
| PartCons | **4.98±0.14** | 4.94±0.37 | **4.97±0.18** | 4.01±1.18 | 6.047 |
| SoftCons | 4.93±0.30 | 4.80±0.77 | 4.91±0.42 | **4.37±0.98** | 5.868 |
| MemSim | 4.93±0.30 | 4.93±0.39 | 4.90±0.41 | 3.61±1.19 | **6.125** |

answers, single-choice answers, and retrieval targets, and report their accuracy. Due to the page limitation, we put the evaluation details in **Appendix B**. According to the results, we find MemDaily can ensure the accuracy of the answers provided for constructed questions.

# 5 Experiments of Benchmark

In this section, we create a benchmark based on the MemDaily dataset, in order to evaluate the memory capability of LLM-based personal assistants. Our benchmark sets various levels of difficulty by introducing different proportions of question-irrelevant daily-life posts.

## 5.1 Experimental Settings

**Levels of Difficulty.** We utilize the MemDaily dataset as the basis of our benchmark. In order to set different levels of difficulty, we collect question-irrelevant posts from social media platforms, and randomly incorporate them into user messages by controlling their proportions. Specifically, we denote MemDaily-vanilla as the vanilla and easiest one without extra additions, and create a series of MemDaily-$\eta$, where we use $\eta$ to represent the inverse percentage of original user messages. Larger $\eta$ indicates a higher level of difficulty in the benchmark. We primarily focus on MemDaily-vanilla and MemDaily-100 as representatives. We also conduct evaluations on MemDaily-10, MemDaily-50, and MemDaily-200, putting their experimental results in **Appendix E**.

**Baselines.** We implement several common memory mechanisms for LLM-based agents according to previous studies [11], including (1) **Full Memory (FullMem)**: saves all previous messages and concatenates them into the prompt for LLM inference. (2) **Recent Memory (ReceMem)**: maintains the most recent $k$ messages and concatenates them into the prompt for LLM inference, also known as short-term memory. (3) **Retrieved Memory (RetrMem)**: stores all previous messages using FAISS [42] and retrieves the top-$k$ relevant messages for inclusion in the prompt for LLM inference, which is commonly used to construct long-term memory. Specifically, we use Llama-160m [43] to transform a message into a 768-dimensional embedding and compute relevance scores using cosine similarity [44]. (4) **None Memory (NonMem)**: does not use memory for LLM inference. Additionally, we include two special baselines for reference: (5) **Noisy Memory (NoisyMem)**: receives only untargeted messages. (6) **Oracle Memory (OracleMem)**: receives only targeted messages. Here, the targeted messages indicate the messages in the ground truth retrieval target. For all methods, we use the open-source GLM-4-9B [45] as the foundational model for its excellent ability in long-context scenarios, and experiments on more backbones can be found in **Appendix F**.

**Metrics.** We propose to evaluate the memory of LLM-based agents from two perspectives: effectiveness and efficiency. Effectiveness refers to the agent's ability to store and utilize factual information. The metrics for effectiveness include: (1) **Accuracy**: The correctness of agents' responses, measured by their ability to answer personal questions based on the factual information from historical user messages. (2) **Recall@5**: The percentage of messages in retrieval target successfully retrieved within the top-5 relevant messages. Efficiency mainly assesses the time cost associated with storing and utilizing information from memory. We use two metrics to evaluate efficiency: (1) **Response Time**: The time taken for an agent to respond after receiving a query, covering the retrieval and utilization processes. (2) **Adaptation Time**: The time required for an agent to store a new message.

## 5.2 Effectiveness of Memory Mechanisms

**Accuracy of factual question-answering.** The results of accuracy are presented in **Table 5**, including the response time for generating answers and adaptation time for storing messages. FullMem and RetrMem demonstrate superior performance compared to other memory mechanisms, achieving high accuracy across both datasets. ReceMem tends to underperform when a large volume of noisy messages is present, as target messages may fall outside the memory window. We observe

Table 5: Results of accuracy for factual question-answering.

| Methods | Simp. | Cond. | Comp. | Aggr. | Post. | Noisy |
|---------|-------|-------|-------|-------|-------|-------|
| **MemDaily-vanilla** | | | | | | |
| FullMem | **0.976±0.022** | **0.982±0.017** | **0.859±0.054** | **0.320±0.079** | **0.848±0.045** | **0.966±0.028** |
| RetrMem | 0.898±0.048 | 0.882±0.040 | 0.771±0.078 | 0.317±0.061 | 0.800±0.054 | 0.786±0.040 |
| ReceMem | 0.832±0.080 | 0.798±0.046 | 0.631±0.069 | 0.257±0.040 | 0.760±0.051 | 0.764±0.042 |
| NonMem | 0.508±0.032 | 0.452±0.059 | 0.157±0.049 | 0.254±0.055 | 0.594±0.073 | 0.380±0.060 |
| NoisyMem | 0.512±0.044 | 0.468±0.054 | 0.204±0.067 | 0.239±0.058 | 0.590±0.045 | 0.388±0.048 |
| OracleMem | 0.966±0.020 | 0.988±0.013 | 0.910±0.032 | 0.376±0.057 | 0.888±0.053 | 0.984±0.017 |
| **MemDaily-100** | | | | | | |
| FullMem | **0.962±0.017** | **0.938±0.033** | 0.586±0.076 | **0.343±0.047** | **0.804±0.041** | **0.872±0.041** |
| RetrMem | 0.892±0.034 | 0.840±0.036 | **0.706±0.074** | 0.320±0.092 | 0.770±0.055 | 0.726±0.052 |
| ReceMem | 0.500±0.063 | 0.442±0.058 | 0.104±0.048 | 0.257±0.054 | 0.600±0.060 | 0.386±0.076 |
| NonMem | 0.508±0.032 | 0.454±0.065 | 0.159±0.052 | 0.252±0.043 | 0.594±0.032 | 0.380±0.057 |
| NoisyMem | 0.458±0.071 | 0.422±0.051 | 0.261±0.068 | 0.283±0.041 | 0.566±0.064 | 0.348±0.044 |
| OracleMem | 0.966±0.020 | 0.988±0.016 | 0.912±0.045 | 0.372±0.062 | 0.888±0.038 | 0.984±0.012 |

Table 6: Results of recall@5 for target message retrieval.

| Methods | Simp. | Cond. | Comp. | Aggr. | Post. | Noisy |
|---------|-------|-------|-------|-------|-------|-------|
| **MemDaily-vanilla** | | | | | | |
| LLM | **0.888±0.025** | **0.851±0.020** | **0.947±0.018** | **0.544±0.021** | **0.800±0.028** | **0.846±0.036** |
| Embedding | 0.735±0.064 | 0.717±0.041 | 0.845±0.022 | 0.515±0.059 | 0.693±0.033 | 0.648±0.018 |
| Recency | 0.514±0.052 | 0.513±0.038 | 0.698±0.034 | 0.237±0.026 | 0.511±0.053 | 0.504±0.047 |
| **MemDaily-100** | | | | | | |
| LLM | 0.612±0.021 | 0.479±0.037 | 0.683±0.036 | 0.290±0.027 | 0.439±0.047 | 0.430±0.059 |
| Embedding | **0.698±0.049** | **0.653±0.061** | **0.778±0.048** | **0.490±0.037** | **0.567±0.042** | **0.543±0.034** |
| Recency | 0.002±0.003 | 0.003±0.004 | 0.002±0.003 | 0.000±0.001 | 0.002±0.003 | $< 0.001$ |

that agents excel with simple, conditional, post-processing, and noisy questions but struggle with comparative and aggregative questions. By comparing with OracleMem, we find the primary difficulty possibly lies in retrieving target messages. Even with accurate retrieval, aggregative questions remain challenging, indicating a potential bottleneck in textual memory. An interesting phenomenon we notice is that NoisyMem shows higher accuracy than NonMem in MemDaily-vanilla but lower accuracy in MemDaily-100. Similarly, FullMem unexpectedly outperforms OracleMem on simple questions in MemDaily. We suspect that LLMs may perform better with memory prompts of medium length, suggesting a potential limitation of textual memory mechanisms for LLM-based agents.

**Recall of target message retrieval.** We implement three retrieval methods to obtain the most relevant messages and compare them with target messages to calculate Recall@5. **Embedding** refers to the retrieval process used in RetrMem. **Recency** considers the most recent $k$ messages as the result. **LLM** directly uses the LLM to respond with the top-$k$ relevant messages. The results are presented in **Table 6**. We find that LLM performs best in short-context scenarios, while Embedding achieves higher recall scores in longer contexts. Additionally, we notice that separating the retrieval and inference stages may exhibit different performances compared with integrating them.

## 5.3 Efficiency of Memory Mechanisms

We put the results in **Appendix C** due to the page limitation. We find that RetrMem consumes the most response time in short-context scenarios, and FullMem also requires more time for inference due to longer memory prompts. However, the response time of FullMem increases significantly faster than that of other methods as the context lengthens. Regarding adaptation time, we observe that RetrMem requires substantially more time because it needs to build indexes in the FAISS system.

# 6 Conclusions and Limitations

In this paper, we propose MemSim, a Bayesian simulator designed to generate reliable datasets for evaluating the memory capability of LLM-based agents. Utilizing MemSim, we generate MemDaily as a dataset in the daily-life scenario, and conduct extensive evaluations to assess its quality. We also provide a benchmark on different memory mechanisms of LLM-based agents and provide further analysis. However, as the very initial study, there are several limitations. Firstly, our work focuses on evaluating the memory capability of LLM-based agents on factual information, but does not address higher-level and abstract information, such as users' hidden preferences. Additionally, our evaluation does not include dialogue forms, which are more complex and challenging to ensure reliability. Besides, preventing the misuse of generated data is also an important consideration. In future works, we aim to address these issues to contribute to the social benefit.

## Acknowledgment

This work is supported in part by National Natural Science Foundation of China (No. 62422215 and No. 62472427), Huawei Innovation Research Programs, Major Innovation & Planning Inter-disciplinary Platform for the "DoubleFirst Class" Initiative, Renmin University of China, Public Computing Cloud, Renmin University of China, fund for building world-class universities (disci-plines) of Renmin University of China.

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

# A Proof in Bayesian Relation Network

## A.1 Proof of Theorem 1

**Theorem 1** (Factorization). *The joint probability distribution of BRNet can be expressed as*

$$P(X_1, X_2, ..., X_{|\mathcal{X}|}) = \prod_{X_t \in \mathcal{X}} P(X_t | par(X_t)),$$

*where $par(X_t)$ denotes the set of parent attributes of $X_t$.*

**Proof.** *Because BRNet is DAG, we can certainly find a topological ordering*

$$O = \begin{bmatrix} o_1, o_2, ..., o_{|\mathcal{X}|} \end{bmatrix}.$$

*Then, we inverse the sequence to get a reversed topologically ordering*

$$\tilde{O} = \begin{bmatrix} \tilde{o}_1, \tilde{o}_2, ..., \tilde{o}_{|\mathcal{X}|} \end{bmatrix}.$$

*Then, we utilize the theorem of conditional probability according to the order $\tilde{O}$, and we have*

$$P(X_1, X_2, ..., X_{|\mathcal{X}|}) = P(X_{\tilde{o}_1} | X_{\tilde{o}_2}, ..., X_{\tilde{o}_{|\mathcal{X}|}}) \cdot P(X_{\tilde{o}_2} | X_{\tilde{o}_3}, ..., X_{\tilde{o}_{|\mathcal{X}|}}) \ldots P(X_{\tilde{o}_{|\mathcal{X}|}}).$$

$$= \prod_{i=1}^{|\mathcal{X}|} P(X_{\tilde{o}_i} | \mathbf{X} \begin{bmatrix} \tilde{o}_{i+1} : \tilde{o}_{|\mathcal{X}|} \end{bmatrix}),$$

*where $\mathbf{X} \begin{bmatrix} \tilde{o}_{i+1} : \tilde{o}_{|\mathcal{X}|} \end{bmatrix}$ means all the variables after $\tilde{o}_{i+1}$ in the reversed topologically ordering, and there are no descendant variables inside. According to **Assumption 1**, we have*

$$P(X_{\tilde{o}_i} | \mathbf{X} \begin{bmatrix} \tilde{o}_{i+1} : \tilde{o}_{|\mathcal{X}|} \end{bmatrix}) = P(X_{\tilde{o}_i} | par(X_{\tilde{o}_i})).$$

*Finally, we rewrite it and obtain*

$$P(X_1, X_2, ..., X_{|\mathcal{X}|}) = \prod_{X_t \in \mathcal{X}} P(X_t | par(X_t)).$$

## A.2 Proof of Theorem 2

**Theorem 2** (Ancestral Sampling). *For BRNet, the result of ancestral sampling is equivalent to that of sampling from the joint probability distribution. Specifically, we have*

$$P(\tilde{x}_1, \tilde{x}_2, ..., \tilde{x}_{|\mathcal{X}|}) = P(x_1, x_2, ..., x_{|\mathcal{X}|}),$$

*where $x_1, x_2, ..., x_{|\mathcal{X}|} \sim P(X_1, X_2, ..., X_{|\mathcal{X}|})$ are sampled from the joint probability distribution.*

**Proof.** *We first calculate the reversed topologically ordering*

$$\tilde{O} = \begin{bmatrix} \tilde{o}_1, \tilde{o}_2, ..., \tilde{o}_{|\mathcal{X}|} \end{bmatrix}.$$

*Then, we have*

$$P(\tilde{x}_1, \tilde{x}_2, ..., \tilde{x}_{|\mathcal{X}|}) = \prod_{i=1}^{|\mathcal{X}|} P(\tilde{x}_{\tilde{o}_i} | \tilde{\mathbf{x}} \begin{bmatrix} \tilde{o}_{i+1} : \tilde{o}_{|\mathcal{X}|} \end{bmatrix})$$

$$= \prod_{i=1}^{|\mathcal{X}|} P(\tilde{x}_{\tilde{o}_i} | par(\tilde{x}_{\tilde{o}_i})).$$

*where $\tilde{\mathbf{x}} \begin{bmatrix} \tilde{o}_{i+1} : \tilde{o}_{|\mathcal{X}|} \end{bmatrix}$ means the values of all the variables after $\tilde{o}_{i+1}$ in the reversed topologically ordering. According to **Assumption 2**, we have*

$$P(\tilde{x}_1, \tilde{x}_2, ..., \tilde{x}_{|\mathcal{X}|}) = \prod_{i=1}^{|\mathcal{X}|} P(x_{\tilde{o}_i} | par(x_{\tilde{o}_i}))$$

$$= P(x_1, x_2, ..., x_{|\mathcal{X}|}).$$

Table 7: Results of the evaluation on questions and answers.

| Question Types | Textual Answers | Single-choice Answers | Retrieval Target |
|---|---|---|---|
| Simple | 100% | 98% | 100% |
| Conditional | 100% | 100% | 100% |
| Comparative | 100% | 100% | 100% |
| Aggregative | 99% | 99% | 100% |
| Post-processing | 100% | 100% | 99% |
| Noisy | 100% | 100% | 100% |
| Average | 99.8% | 99.5% | 99.8% |

# B   Evaluation on Questions and Answers

The primary challenge for constructing a reliable dataset is ensuring the accuracy of ground truths for the constructed questions. To assess the reliability of MemDaily, we sample approximately 20% of all the trajectories in MemDaily and employ human evaluators to verify the correctness of their ground truths. Specifically, the evaluators are required to examine three parts of the ground truths: textual answers, single-choice answers, and retrieval targets, and report their accuracy. The guidelines of human evaluators are provided in **Appendix H.3**.

**Metrics.** The accuracy of textual answers assesses whether an answer correctly responds to the question based on the user messages within the same trajectory. The accuracy of single-choice answers indicates whether the ground truth choice is the sole correct answer for the question, given the user messages, while other choices are incorrect. The accuracy of retrieval targets evaluates whether the messages of the retrieval target are sufficient and necessary to answer the question.

**Results.** As shown in **Table 7**, MemDaily significantly ensures the accuracy of the answers provided for constructed questions. In the few instances where accuracy is compromised, it is attributed to the rewriting process by LLMs, which occasionally leads to information deviation. The results also demonstrate that MemSim can effectively mitigate the impact of LLM hallucinations on factual information, addressing a critical challenge in generating reliable questions and answers for memory evaluation. Another baseline method that directly generates answers through LLMs based on targeted user messages and questions performs much lower reliability. We implement this method and present the results as *OracleMem* in our constructed benchmarks in **Section 5.2**.

# C Benchmark on the Efficiency of Memory Mechanisms

The results of efficiency are presented in **Table 8** and **Table 9**.

Table 8: Results of response time for generating answers (seconds per query).

| Methods | Simp. | Cond. | Comp. | Aggr. | Post. | Noisy |
|---|---|---|---|---|---|---|
| **MemDaily-vanilla** | | | | | | |
| FullMem | 0.139±0.001 | 0.141±0.001 | 0.132±0.001 | 0.154±0.002 | 0.152±0.002 | 0.150±0.003 |
| RetrMem | 0.290±0.007 | 0.277±0.007 | 0.267±0.009 | 0.236±0.009 | 0.257±0.004 | 0.284±0.007 |
| ReceMem | 0.126±0.001 | 0.127±0.001 | 0.125±0.000 | 0.125±0.001 | 0.135±0.001 | 0.134±0.001 |
| NonMem | 0.118±0.000 | 0.119±0.000 | 0.118±0.000 | 0.118±0.000 | 0.121±0.001 | 0.121±0.000 |
| NoisyMem | 0.118±0.000 | 0.119±0.000 | 0.118±0.001 | 0.118±0.000 | 0.121±0.001 | 0.121±0.000 |
| OracleMem | 0.122±0.001 | 0.122±0.001 | 0.122±0.000 | 0.131±0.001 | 0.129±0.002 | 0.128±0.001 |
| **MemDaily-100** | | | | | | |
| FullMem | 1.632±0.097 | 1.648±0.101 | 1.196±0.077 | 2.522±0.129 | 1.782±0.136 | 1.799±0.102 |
| RetrMem | 0.207±0.020 | 0.223±0.005 | 0.228±0.011 | 0.205±0.008 | 0.228±0.029 | 0.284±0.022 |
| ReceMem | 0.120±0.000 | 0.125±0.008 | 0.121±0.001 | 0.120±0.000 | 0.125±0.001 | 0.124±0.001 |
| NonMem | 0.119±0.001 | 0.119±0.000 | 0.119±0.000 | 0.119±0.001 | 0.123±0.000 | 0.122±0.001 |
| NoisyMem | 1.578±0.124 | 1.591±0.187 | 1.153±0.073 | 2.424±0.138 | 1.717±0.095 | 1.735±0.158 |
| OracleMem | 0.122±0.001 | 0.123±0.001 | 0.123±0.001 | 0.132±0.001 | 0.130±0.001 | 0.129±0.001 |

Table 9: Results of adaptation time for storing messages (seconds per message).

| Methods | Simp. | Cond. | Comp. | Aggr. | Post. | Noisy |
|---|---|---|---|---|---|---|
| **MemDaily-vanilla** | | | | | | |
| RetrMem | 0.222±0.009 | 0.182±0.004 | 0.151±0.009 | 0.136±0.010 | 0.133±0.004 | 0.112±0.005 |
| Others | < 0.001 | < 0.001 | < 0.001 | < 0.001 | < 0.001 | < 0.001 |
| **MemDaily-100** | | | | | | |
| RetrMem | 0.064±0.008 | 0.072±0.004 | 0.066±0.007 | 0.064±0.006 | 0.056±0.002 | 0.066±0.005 |
| Others | <0.001 | <0.001 | <0.001 | <0.001 | <0.001 | <0.001 |

# D Extensive Evaluation on User Messages by GPT-4o

We also let GPT-4o score on user messages as a reference, and the results are shown in **Table 10**.

Table 10: Results of evaluation on user messages by GPT-4o.

| Methods | F-GPT | R-GPT | N-GPT | I-GPT |
|---|---|---|---|---|
| ZeroCons | 4.04 | 4.80 | 4.60 | 3.04 |
| PartCons | **4.28** | 4.88 | 4.80 | **4.28** |
| SoftCons | 4.20 | **5.00** | **5.00** | 3.96 |
| MemSim | 4.04 | 4.84 | 4.68 | 3.60 |

# E Extensive Benchmark on More Composite Datasets

## E.1 Results on MemDaily-10

The results of accuracy are shown in **Table 11**. The results of recall@5 are shown in **Table 12**. The results of response time are shown in **Table 13**. The results of adaptation time are shown in **Table 14**.

Table 11: Results of accuracy on MemDaily-10.

| Methods | Simp. | Cond. | Comp. | Aggr. | Post. | Noisy |
|---|---|---|---|---|---|---|
| FullMem | **0.962±0.040** | **0.966±0.028** | 0.665±0.058 | 0.243±0.072 | **0.810±0.036** | **0.922±0.029** |
| RetrMem | 0.896±0.033 | 0.882±0.047 | **0.759±0.068** | **0.315±0.045** | 0.782±0.065 | 0.764±0.053 |
| ReceMem | 0.534±0.047 | 0.482±0.064 | 0.147±0.049 | 0.248±0.067 | 0.604±0.088 | 0.430±0.048 |
| NonMem | 0.510±0.090 | 0.450±0.078 | 0.159±0.041 | 0.254±0.065 | 0.594±0.032 | 0.380±0.057 |
| NoisyMem | 0.428±0.068 | 0.402±0.059 | 0.169±0.046 | 0.280±0.046 | 0.584±0.090 | 0.350±0.077 |
| OracleMem | 0.966±0.022 | 0.988±0.010 | 0.910±0.031 | 0.372±0.037 | 0.888±0.030 | 0.888±0.030 |

Table 12: Results of recall@5 on MemDaily-10.

| Methods | Simp. | Cond. | Comp. | Aggr. | Post. | Noisy |
|---|---|---|---|---|---|---|
| LLM | **0.794±0.035** | **0.872±0.019** | **0.518±0.027** | **0.732±0.036** | **0.756±0.038** | **0.846±0.036** |
| Embedding | 0.704±0.039 | 0.833±0.026 | 0.506±0.052 | 0.643±0.043 | 0.609±0.027 | 0.648±0.018 |
| Recency | 0.032±0.017 | 0.011±0.010 | 0.013±0.011 | 0.030±0.012 | 0.009±0.007 | 0.504±0.047 |

Table 13: Results of response time on MemDaily-10 (seconds per query).

| Methods | Simp. | Cond. | Comp. | Aggr. | Post. | Noisy |
|---|---|---|---|---|---|---|
| FullMem | 0.243±0.008 | 0.243±0.008 | 0.208±0.003 | 0.306±0.008 | 0.263±0.006 | 0.262±0.010 |
| RetrMem | 0.213±0.002 | 0.230±0.005 | 0.246±0.008 | 0.212±0.002 | 0.240±0.004 | 0.292±0.014 |
| ReceMem | 0.120±0.000 | 0.121±0.000 | 0.120±0.000 | 0.119±0.002 | 0.126±0.001 | 0.124±0.001 |
| NonMem | **0.119±0.000** | **0.119±0.001** | **0.119±0.000** | **0.117±0.002** | **0.122±0.000** | **0.119±0.002** |
| NoisyMem | 0.205±0.005 | 0.207±0.007 | 0.181±0.004 | 0.253±0.010 | 0.223±0.005 | 0.222±0.006 |
| OracleMem | 0.121±0.001 | 0.123±0.001 | 0.122±0.000 | 0.131±0.001 | 0.130±0.001 | 0.128±0.001 |

Table 14: Results of adaptation time on MemDaily-10 (seconds per message).

| Methods | Simp. | Cond. | Comp. | Aggr. | Post. | Noisy |
|---|---|---|---|---|---|---|
| FullMem | < 0.001 | < 0.001 | < 0.001 | < 0.001 | < 0.001 | < 0.001 |
| RetrMem | 0.073±0.003 | 0.079±0.006 | 0.084±0.006 | 0.069±0.003 | 0.073±0.003 | 0.075±0.006 |
| ReceMem | < 0.001 | < 0.001 | < 0.001 | < 0.001 | < 0.001 | < 0.001 |
| NonMem | < 0.001 | < 0.001 | < 0.001 | < 0.001 | < 0.001 | < 0.001 |
| NoisyMem | < 0.001 | < 0.001 | < 0.001 | < 0.001 | < 0.001 | < 0.001 |
| OracleMem | < 0.001 | < 0.001 | < 0.001 | < 0.001 | < 0.001 | < 0.001 |

## E.2    Results of MemDaily-50

The results of accuracy are shown in **Table 15**. The results of recall@5 are shown in **Table 16**. The results of response time are shown in **Table 17**. The results of adaptation time are shown in **Table 18**.

Table 15: Results of accuracy on MemDaily-50.

| Methods | Simp. | Cond. | Comp. | Aggr. | Post. | Noisy |
|---|---|---|---|---|---|---|
| FullMem | **0.962±0.027** | **0.948±0.020** | 0.602±0.065 | 0.296±0.072 | **0.802±0.046** | **0.880±0.041** |
| RetrMem | 0.886±0.035 | 0.864±0.037 | **0.724±0.062** | **0.320±0.071** | 0.780±0.059 | 0.748±0.049 |
| ReceMem | 0.508±0.042 | 0.434±0.052 | 0.108±0.044 | 0.237±0.054 | 0.588±0.066 | 0.376±0.099 |
| NonMem | 0.510±0.061 | 0.452±0.055 | 0.159±0.039 | 0.254±0.066 | 0.594±0.078 | 0.380±0.055 |
| NoisyMem | 0.454±0.040 | 0.416±0.083 | 0.229±0.071 | 0.272±0.073 | 0.568±0.078 | 0.360±0.084 |
| OracleMem | 0.966±0.025 | 0.988±0.010 | 0.910±0.053 | 0.376±0.042 | 0.888±0.032 | 0.984±0.012 |

Table 16: Results of recall@5 on MemDaily-50.

| Methods | Simp. | Cond. | Comp. | Aggr. | Post. | Noisy |
|---|---|---|---|---|---|---|
| LLM | **0.725±0.047** | 0.640±0.053 | 0.773±0.018 | 0.373±0.031 | 0.591±0.039 | 0.561±0.050 |
| Embedding | 0.710±0.041 | **0.674±0.021** | **0.790±0.037** | **0.497±0.039** | **0.591±0.037** | **0.564±0.053** |
| Recency | 0.011±0.009 | 0.005±0.004 | 0.006±0.006 | 0.001±0.002 | 0.003±0.004 | 0.001±0.003 |

Table 17: Results of response time on MemDaily-50 (seconds per query).

| Methods | Simp. | Cond. | Comp. | Aggr. | Post. | Noisy |
|---|---|---|---|---|---|---|
| FullMem | 0.776±0.031 | 0.783±0.067 | 0.596±0.021 | 1.134±0.054 | 0.841±0.032 | 0.847±0.062 |
| RetrMem | 0.203±0.003 | 0.206±0.004 | 0.215±0.004 | 0.204±0.003 | 0.229±0.005 | 0.324±0.020 |
| ReceMem | 0.120±0.001 | 0.121±0.002 | 0.118±0.000 | 0.118±0.001 | 0.123±0.002 | 0.123±0.001 |
| NonMem | **0.118±0.001** | **0.118±0.002** | **0.117±0.002** | **0.118±0.001** | **0.121±0.001** | **0.119±0.001** |
| NoisyMem | 0.728±0.037 | 0.737±0.041 | 0.562±0.027 | 1.060±0.055 | 0.787±0.028 | 0.794±0.058 |
| OracleMem | 0.121±0.001 | 0.122±0.001 | 0.121±0.001 | 0.131±0.001 | 0.129±0.001 | 0.128±0.001 |

Table 18: Results of adaptation time on MemDaily-50 (seconds per message).

| Methods | Simp. | Cond. | Comp. | Aggr. | Post. | Noisy |
|---|---|---|---|---|---|---|
| FullMem | < 0.001 | < 0.001 | < 0.001 | < 0.001 | < 0.001 | < 0.001 |
| RetrMem | 0.059±0.001 | 0.057±0.003 | 0.057±0.004 | 0.060±0.003 | 0.062±0.003 | 0.089±0.005 |
| ReceMem | < 0.001 | < 0.001 | < 0.001 | < 0.001 | < 0.001 | < 0.001 |
| NonMem | < 0.001 | < 0.001 | < 0.001 | < 0.001 | < 0.001 | < 0.001 |
| NoisyMem | < 0.001 | < 0.001 | < 0.001 | < 0.001 | < 0.001 | < 0.001 |
| OracleMem | < 0.001 | < 0.001 | < 0.001 | < 0.001 | < 0.001 | < 0.001 |

## E.3 Results of MemDaily-200

The results of accuracy are shown in **Table 19**. The results of recall@5 are shown in **Table 20**. The results of response time are shown in **Table 21**. The results of adaptation time are shown in **Table 22**.

Table 19: Results of accuracy on MemDaily-200.

| Methods | Simp. | Cond. | Comp. | Aggr. | Post. | Noisy |
|---|---|---|---|---|---|---|
| FullMem | **0.932±0.040** | **0.932±0.036** | 0.563±0.061 | 0.309±0.056 | **0.782±0.045** | **0.866±0.044** |
| RetrMem | 0.874±0.052 | 0.844±0.034 | **0.704±0.061** | **0.315±0.065** | 0.766±0.046 | 0.714±0.052 |
| ReceMem | 0.486±0.046 | 0.420±0.057 | 0.114±0.036 | 0.272±0.054 | 0.570±0.055 | 0.366±0.051 |
| NonMem | 0.470±0.057 | 0.454±0.077 | 0.157±0.045 | 0.257±0.069 | 0.592±0.082 | 0.380±0.048 |
| NoisyMem | 0.398±0.052 | 0.398±0.068 | 0.282±0.058 | 0.276±0.068 | 0.564±0.037 | 0.350±0.035 |
| OracleMem | 0.990±0.013 | 0.988±0.013 | 0.910±0.034 | 0.374±0.063 | 0.888±0.056 | 0.984±0.012 |

Table 20: Results of recall@5 on MemDaily-200.

| Methods | Simp. | Cond. | Comp. | Aggr. | Post. | Noisy |
|---|---|---|---|---|---|---|
| LLM | 0.457±0.066 | 0.356±0.051 | 0.556±0.035 | 0.176±0.022 | 0.342±0.048 | 0.322±0.043 |
| Embedding | **0.674±0.052** | **0.641±0.044** | **0.753±0.036** | **0.484±0.050** | **0.544±0.054** | **0.508±0.052** |
| Recency | 0.001±0.003 | 0.001±0.002 | 0.001±0.002 | 0.000±0.001 | 0.001±0.003 | 0.000±0.000 |

Table 21: Results of response time on MemDaily-200 (seconds per query).

| Methods | Simp. | Cond. | Comp. | Aggr. | Post. | Noisy |
|---|---|---|---|---|---|---|
| FullMem | 4.028±0.161 | 3.914±0.213 | 2.697±0.100 | 6.365±0.374 | 4.252±0.328 | 4.307±0.283 |
| RetrMem | 0.236±0.023 | 0.241±0.018 | 0.238±0.024 | 0.585±0.230 | 1.012±0.690 | 1.252±0.427 |
| ReceMem | **0.130±0.002** | 0.120±0.002 | **0.118±0.001** | 0.119±0.001 | 0.124±0.001 | 0.123±0.001 |
| NonMem | 0.139±0.006 | **0.119±0.001** | 0.119±0.001 | **0.117±0.001** | **0.121±0.001** | **0.121±0.001** |
| NoisyMem | 3.947±0.209 | 3.832±0.203 | 2.637±0.118 | 6.221±0.325 | 4.158±0.226 | 4.214±0.288 |
| OracleMem | 0.141±0.003 | 0.122±0.001 | 0.121±0.001 | 0.131±0.002 | 0.128±0.002 | 0.128±0.001 |

Table 22: Results of adaptation time on MemDaily-200 (seconds per message).

| Methods | Simp. | Cond. | Comp. | Aggr. | Post. | Noisy |
|---|---|---|---|---|---|---|
| FullMem | <0.001 | <0.001 | <0.001 | <0.001 | <0.001 | <0.001 |
| RetrMem | 0.080±0.011 | 0.080±0.013 | 0.080±0.010 | 0.220±0.076 | 0.264±0.089 | 0.420±0.120 |
| ReceMem | <0.001 | <0.001 | <0.001 | <0.001 | <0.001 | <0.001 |
| NonMem | <0.001 | <0.001 | <0.001 | <0.001 | <0.001 | <0.001 |
| NoisyMem | <0.001 | <0.001 | <0.001 | <0.001 | <0.001 | <0.001 |
| OracleMem | <0.001 | <0.001 | <0.001 | <0.001 | <0.001 | <0.001 |

# F Supplementary Experiments on More Backbones

To better compare the performance across different models, we conduct supplementary experiments with Qwen2.5-7B on MemDaily-vanilla. The results are shown in **Table 23** and **Table 24**.

Table 23: Results of accuracy with Qwen2.5-7B on MemDaily-vanilla.

| Methods | Simp. | Cond. | Comp. | Aggr. | Post. | Noisy |
|---|---|---|---|---|---|---|
| FullMem | 0.980±0.008 | 0.974±0.017 | 0.876±0.029 | 0.312±0.045 | 0.853±0.003 | 0.966±0.006 |
| RetrMem | 0.880±0.005 | 0.878±0.010 | 0.709±0.019 | 0.229±0.027 | 0.787±0.024 | 0.799±0.008 |
| ReceMem | 0.809±0.023 | 0.797±0.010 | 0.512±0.026 | 0.145±0.012 | 0.739±0.033 | 0.743±0.046 |
| NonMem | 0.526±0.017 | 0.424±0.027 | 0.183±0.005 | 0.255±0.013 | 0.629±0.033 | 0.376±0.010 |
| NoisyMem | 0.512±0.017 | 0.424±0.050 | 0.262±0.017 | 0.301±0.017 | 0.637±0.032 | 0.373±0.020 |
| OracleMem | 0.982±0.005 | 0.976±0.005 | 0.909±0.023 | 0.450±0.089 | 0.880±0.020 | 0.978±0.008 |

Table 24: Results of recall with Qwen2.5-7B on MemDaily-vanilla.

| Methods | Simp. | Cond. | Comp. | Aggr. | Post. | Noisy |
|---|---|---|---|---|---|---|
| FullMem | 0.939±0.006 | 0.884±0.007 | 0.918±0.009 | 0.601±0.016 | 0.863±0.003 | 0.872±0.019 |
| RetrMem | 0.726±0.013 | 0.717±0.023 | 0.846±0.008 | 0.516±0.029 | 0.691±0.014 | 0.649±0.014 |
| ReceMem | 0.596±0.001 | 0.524±0.025 | 0.700±0.010 | 0.223±0.012 | 0.501±0.023 | 0.503±0.031 |
| NonMem | 0.319±0.013 | 0.391±0.019 | 0.414±0.048 | 0.183±0.008 | 0.319±0.006 | 0.562±0.017 |
| NoisyMem | 0.575±0.025 | 0.525±0.014 | 0.725±0.012 | 0.454±0.007 | 0.487±0.025 | 0.660±0.019 |
| OracleMem | 0.406±0.021 | 0.609±0.035 | 0.782±0.016 | 0.562±0.025 | 0.605±0.009 | 0.585±0.023 |

Based on the experimental results for Qwen2.5-7B, a clear performance hierarchy emerges across different memory mechanisms, with FullMem and OracleMem consistently achieving the highest accuracy scores, followed by moderate performance from RetrMem and ReceMem, while NonMem and NoisyMem demonstrate substantially degraded performance (0.183-0.637 range). The results reveal task-specific challenges, particularly evident in the Aggregation task where all methods show reduced accuracy, and demonstrate a trade-off between accuracy and recall metrics, with some methods like OracleMem showing lower recall despite high accuracy, suggesting potential overfitting or conservative prediction behavior. This is consistent with the conclusions drawn from GLM-4-9B.

# G Case Studies

In this section, we present several case studies to illustrate the effectiveness of the data generated by MemDaily. First, we will display the hierarchical user profiles generated from BRNet. Next, we will present examples of user messages created by our method. Finally, we will provide examples of questions and answers for each type.

## G.1 Case Study on Generated User Profiles

In MemDaily, we incorporate 11 entities that cover 7 types, with 73 attributes of them. The summary of entities and attributes of MemDaily are provided in **Table 25**.

We introduce prior knowledge as several rules according to our scenarios to constrain among attributes. For example, a relative role is highly possible to share the same hometown with the user, because they are likely to come from the same place. All of these constraints are expressed in BRNet with causal relations. We generate 50 graphical user profiles and conduct observations, finding that most profiles align well with real-world users without contradictions.

Here is a case of user profiles, and we translate them into English for better demonstration:

---

**An example of Generated User Profiles**

**User Profiles:**
(Gender) Male; (Name) Qiang Wang; (Age) 38; (Height) 166cm; (Birthday) December 1st.; (Hometown) Beijing; (Workplace) Shenzhen, Guangdong; (Education) High School; (Occupation) Bank Teller; (Position) Head Teller; (Company) Huayin Financial Service Center; (Hobbies) Model Making; (Personality) Outgoing; (Phone) 13420824898; (Email) wangqiang1201@huayinfinance.com; (ID Number) 640168198612016598; (Passport Number) NZ0448096; (Bank Card Number) 6222022612177604; (Driver's License Number) 640168198612012730;

**College Role 1:**
(Gender) Female; (Relationship) Supervisor; (Name) Yalin Zhao; (Age) 44; (Height) 165cm; (Birthday) Febrary 5th.; (Hometown) Chongqing; (Workplace) Shenzhen, Guangdong; (Education) High School; (Occupation) Bank Teller; (Position) Bank Manager; (Company) Huayin Financial Service Center; (Hobbies) Sports; (Personality) Patient; (Phone) 13651039007; (Email) zhaoyalin0205@szfinancecenter.com;

**College Role 2:**
(Gender) Male; (Relationship) Colleague; (Name) Zhihong Sun; (Age) 39; (Height) 164cm; (Birthday) April 24th.; (Hometown) Chengdu, Sichuan; (Workplace) Shenzhen, Guangdong; (Education) High School; (Occupation) Bank Teller; (Position) Senior Teller; (Company) Huayin Financial Service Center; (Hobbies) Attending concerts; (Personality) Enthusiastic; (Phone) 15391721618; (Email) sunzhihong0421@huayinfinance.com;

**Relative Role 1:**
(Gender) Male; (Relationship) Cousin; (Name) Wei Zhang; (Age) 36; (Height) 169cm; (Birthday) July 15th.; (Hometown) Beijing; (Workplace) Hangzhou, Zhejiang; (Education) Doctor; (Occupation) Doctor; (Position) Chief Physician; (Company) West Lake Hospital; (Hobbies) Playing Video Games; (Personality) Patient; (Phone) 13225162475; (Email) zhangwei0715@westlakehospital.com;

**Relative Role 2:**
(Gender) Female; (Relationship) Cousin; (Name) Tingting Li; (Age) 36; (Height) 164cm; (Birthday) June 23rd.; (Hometown) Beijing; (Workplace) Shanghai; (Education) Master; (Occupation) Teacher; (Position) Middle School Language Teacher; (Company) Pudong No.1 Middle School; (Hobbies) Yoga; (Personality) Patient; (Phone) 13401551341; (Email) litingting0623@pdxzyz.com;

**Work Event 1:**
(Type) Job Fair; (Content) Job Fair for Bank Teller Supervisors in the Shenzhen area, sharing professional experience, recruiting talented individuals, and jointly creating a brilliant future for the banking industry.; (Location) Shenzhen, Guangdong; (Time) At 7 PM on the Sunday after next; (Title) Bank Teller Job Fair; (Scale) Around 500 People; (Duration) Eight Weeks;

---

> **An example of Generated User Profile**
>
> **Work Event 2:**
> (Type) Academic Exchange Conference; (Content) Discuss the development trends of financial technology, share experiences in innovative banking services, and promote communication and cooperation among industry elites.; (Location) Beijing; (Time) Next Saturday at 2 PM; (Title) Financial Technology Elite Forum; (Scale) Around 3000 People; (Duration) Seven days;
> **Entertainment Event 1:**
> (Type) Art Exhibition; (Content) Displaying selected model works, exchanging making techniques, experiencing creative handicrafts, and feeling the charm of art.; (Location) Beijing; (Time) At 7 PM on the coming Monday; (Title) Model Art Feast; (Scale) Around 900 People; (Duration) Seven Days; (Relationship) Live;
> **Entertainment Event 2:**
> (Type) Outdoor Hiking; (Content) Conduct outdoor hiking activities, combined with model making, taking natural scenery along the way, creating outdoor landscape models, and sharing modeling techniques.; (Location) Guangdong, Shenzhen; (Time) The Wednesday evening at seven in two weeks; (Title) Outdoor Hiking Model Creation Journey; (Scale) Around 900 People; (Duration) Seven Days; (Relationship) Eight weeks;
> **Place:**
> (Type) Residential Community; (Name) Oasis Home; (Comment) Oasis Home is really a nice place to live, with a high green coverage rate and a beautiful environment. It's especially great to walk and relax here after work every day. However, the commercial facilities are slightly lacking, and it would be perfect if there were more convenience stores and restaurants.; (Relationship) Use;
> **Item:**
> (Type) Sports Shoes; (Name) ASICS Gel-Kayano 26; (Comment) These ASICS Gel-Kayano 26 shoes are really great, especially for their stability and support, which is perfect for standing work for long periods. Wearing them, my feet feel much more comfortable. However, it would be perfect if they had better breathability.;

From the case profile in MemDaily, we find that our generated user profiles can greatly align with that in real-world scenarios.

Table 25: Summary of entities and attributes of MemDaily.

| Entity | Attribute | Entity | Attribute |
|---|---|---|---|
| User (self) | Gender
Name
Age
Height
Birthday
Hometown
Workplace
Education
Occupation
Position
Company
Hobbies
Personality
Phone | Relative Roles | Name
Age
Height
Birthday
Hometown
Workplace
Education
Occupation
Position
Company
Hobbies
Personality
Phone
Email |
| | Email
ID Number
Passport Number
Bank Card Number
Driver's License Number | Work Events | Type
Content
Location
Time
Title
Scale
Duration |
| College Roles | Gender
Relationship
Name
Age
Height
Birthday
Hometown
Workplace
Education
Occupation
Position
Company
Hobbies
Personality
Phone
Email | Entertainment Events | Type
Content
Location
Time
Title
Scale
Duration |
| | | Places | Relationship
Type
Name
Comment |
| Relative Roles | Gender
Relationship | Items | Relationship
Type
Name
Comment |
| | | Total (7) | Total (73) |

## G.2 Case Study on User Messages

Based on the generated user profiles, we further generate user messages without inside contradictory according to **Section 3.3**. Here is a case of message list (translated into English) in **Table 26**.

Table 26: A case of user messages.

| Index | Message | Time | Place |
|---|---|---|---|
| 0 | My colleague's email is sunzhihong0421@huayinfinance.com. | April 1, 2024, Monday, 08:07 | Guangdong Shenzhen |
| 1 | My colleague really likes to attend concerts. | April 2, 2024, Tuesday, 07:01 | Guangdong Shenzhen |
| 2 | My colleague's phone number is 15391721618. | April 2, 2024, Tuesday, 08:23 | Guangdong Shenzhen |
| 3 | My colleague's birthday is on April 21st. | April 2, 2024, Tuesday, 17:02 | Guangdong Shenzhen |
| 4 | My colleague's name is Zhihong Sun. | April 3, 2024, Wednesday, 07:49 | Guangdong Shenzhen |
| 5 | Wei Zhang's email address is zhangwei0715@westlakehospital.com. | April 3, 2024, Wednesday, 19:07 | Guangdong Shenzhen |
| 6 | Tingting Li's email address is litingting0623@pdxzyz.com. | April 4, 2024, Thursday, 07:16 | Guangdong Shenzhen |
| 7 | Yalin Zhao's email address is zhaoyalin0205@szfinancecenter.com. | April 4, 2024, Thursday, 13:38 | Guangdong Shenzhen |
| 8 | I am going to attend the bank teller job fair. | April 5, 2024, Friday, 16:21 | Guangdong Shenzhen |
| 9 | The time for the bank teller job fair is at seven o'clock in the evening on the next Sunday. | April 6, 2024, Saturday, 07:18 | Guangdong Shenzhen |
| 10 | The location of the bank teller job fair is in Guangdong Shenzhen. | April 6, 2024, Saturday, 16:58 | Guangdong Shenzhen |
| 11 | The main content of the bank teller job fair is the job fair: Shenzhen area bank head teller, sharing professional experience, recruiting talent, creating a brilliant bank career together. | April 7, 2024, Sunday, 07:21 | Guangdong Shenzhen |
| 12 | The time for the Financial Technology Elite Forum is at two o'clock in the afternoon next Saturday. | April 7, 2024, Sunday, 21:33 | Guangdong Shenzhen |
| 13 | The time for the Model Art Banquet is at seven o'clock in the evening next Monday. | April 8, 2024, Monday, 12:45 | Guangdong Shenzhen |
| 14 | The time for the Outdoor Hiking Model Creation Journey is at seven o'clock in the evening on the next Wednesday. | April 9, 2024, Tuesday, 07:36 | Guangdong Shenzhen |

By utilizing our mechanisms, we can ensure that there is no contradiction among user messages. We further demonstrate the list of hints that correspond to the above messages in **Table 27**.

Table 27: A case of the hint list.

| Index | Entity | Attribute | Value |
|---|---|---|---|
| 0 | Colleague Role 2 | Email | sunzhihong0421@huayinfinance.com |
| 1 | Colleague Role 2 | Hobbies | Attend Concerts |
| 2 | Colleague Role 2 | Phone | 15391721618 |
| 3 | Colleague Role 2 | Birthday | April 21st |
| 4 | Colleague Role 2 | Name | Zhihong Sun |
| 5 | Relative Role 1 | Email | zhangwei0715@westlakehospital.com |
| 6 | Relative Role 2 | Email | litingting0623@pdxzyz.com |
| 7 | Colleague Role 1 | Email | zhaoyalin0205@szfinancecenter.com |
| 8 | Work Event 1 | Title | Bank Teller Job Fair; |
| 9 | Work Event 1 | Time | At 7 PM on the Sunday after next |
| 10 | Work Event 1 | Location | Shenzhen, Guangdong |
| 11 | Work Event 1 | Content | Job Fair for Bank Teller Supervisors in the Shenzhen area, sharing professional experience, recruiting talented individuals, and jointly creating a brilliant future for the banking industry |
| 12 | Work Event 2 | Time | Next Saturday at 2 PM; (Title) Financial Technology Elite Forum |
| 13 | Entertainment Event 1 | Time | At 7 PM on the coming Monday |
| 14 | Entertainment Event 2 | Time | The Wednesday evening at seven in two weeks |

### G.3 Case Study on Questions and Answers

In this section, we will show the cases of questions and answers of different types. We leave out the time and place of each message in this section, where they do not influence the QA in these cases. We have translated all texts into English for better demonstration.

**Simple** *(Simp.)* Simple QAs in single-hop.

---

**A Case of Simple Questions and Answers**

**Messages:**
[0] My cousin's email address is zhangwei0715@westlakehospital.com.
[1] My cousin works in Hangzhou, Zhejiang.
[2] My cousin is 169 cm tall.
[3] My cousin is from Beijing.
[4] My cousin is 36 years old this year.
[5] My sister is of her 36 age as well.
[6] My boss is 44 years old.
[7] My colleague is 39 years old this year.
**Question:**
How old is my cousin now?
**Answer(Text):**
36 years old.
**Choices:**
A. 35 years old.
B. 37 years old.
C. 34 years old.
D. 36 years old.
**Answer(Choice):** D
**Answer(Retrieval):** [4]
**Time:** April 5, 2024, Friday 07:54

---

**Conditional** *(Cond.)* Conditional QAs in multi-hop.

---

**A Case of Conditional Questions and Answers**

**Messages:**
[0] My boss only has a high school education.
[1] My boss works as a bank teller.
[2] My boss's contact phone number is 13651039007.
[3] My boss is 165cm tall.
[4] My boss works in Shenzhen, the one in Guangdong.
[5] My cousin works in Hangzhou, Zhejiang.
[6] My cousin works in Shanghai.
[7] My colleague works in Shenzhen, in Guangdong.
**Question:**
Where does the person with only a high school education work now?
**Answer(Text):**
Shenzhen, Guangdong.
**Choices:**
A. Zhuhai, Guangdong.
B. Shenzhen, Guangdong.
C. Shenzhen, Guangzhou.
D. Xiamen, Fujian.
**Answer(Choice):** B
**Answer(Retrieval):** [0, 4]
**Time:** April 6, 2024, Saturday 07:24

---

**Comparative** *(Comp.)* Comparative QAs in multi-hop.

> ### A Case of Comparative Questions and Answers
>
> **Messages:**
> [0] Yalin Zhao is my boss, who is 44 years old.
> [1] Wei Zhang is my cousin, and he is 36 years old.
> [2] Tingting Li is my cousin, and she is 36 years old.
> [3] Zhihong Sun is my colleague, and he is 39 years old.
> **Question:**
> Who is older, Yalin Zhao or Wei Zhang?
> **Answer(Text):**
> Yalin Zhao.
> **Choices:**
> A. Yalin Zhao.
> B. Wei Zhang.
> C. Both are the same age.
> D. Neither is correct.
> **Answer(Choice):** A
> **Answer(Retrieval):** [0, 1]
> **Time:** April 3, 2024, Wednesday 14:38

**Aggregative** *(Aggr.)* Aggregative QAs in multi-hop.

> ### A Case of Aggregative Questions and Answers
>
> **Messages:**
> [0] Wei Zhang is my cousin, and his educational background is a Ph.D.
> [1] Tingting Li is my cousin, and her educational background is a master's degree.
> [2] Yalin Zhao is my boss, and her educational background is high school.
> [3] Zhihong Sun is my colleague, and his educational background is high school.
> [4] Wei Zhang is my cousin, and his hometown is Beijing.
> [5] Tingting Li is my cousin, and her hometown is Beijing.
> [6] Yalin Zhao is my boss, and her hometown is Chongqing.
> [7] Zhihong Sun is my colleague, and his hometown is Chengdu, Sichuan.
> **Question:**
> How many people have an educational background of high school or below?
> **Answer(Text):**
> 2 people.
> **Choices:**
> A. 3 people.
> B. 1 people.
> C. 4 people.
> D. 2 people.
> **Answer(Choice):** D
> **Answer(Retrieval):** [0, 1, 2, 3]
> **Time:** April 5, 2024, Friday 07:27

**Post-processing** *(Post.)* Multi-hop QAs that requires extra reasoning steps.

---

### A Case of Post-processing Questions and Answers

**Messages:**
[0] My cousin works in Hangzhou, Zhejiang.
[1] My cousin likes to play video games.
[2] My cousin's birthday is July 15th.
[3] My cousin's email address is zhangwei0715@westlakehospital.com.
[4] My cousin's phone number is 13225162475.
[5] Tingting Li works in Shanghai.
[6] Yalin Zhao works in Shenzhen, Guangdong.
[7] Zhihong Sun works in Shenzhen, Guangdong.

**Question:**
Which of the following descriptions matches the work location of the person whose birthday is July 15th?

**Answer(Text):**
A city with beautiful West Lake scenery and a developed internet industry.

**Choices:**
A. Capital, political and cultural center.
B. International metropolis, economic and financial center
C. A city with beautiful West Lake scenery and a developed internet industry.
D. Special economic zone, an important city for technological innovation.

**Answer(Choice):** C
**Answer(Retrieval):** [0, 2]
**Time:** April 6, 2024, Saturday 07:51

---

**Noisy** *(Nois.)* Multi-hop QAs that add extra noise in questions.

---

### A Case of Noisy Questions and Answers

**Messages:**
[0] My boss is 44 years old this year.
[1] My boss is the head of a bank.
[2] My boss works in Shenzhen, Guangdong.
[3] My boss really likes sports.
[4] My boss's phone number is 13651039007.
[5] My cousin really likes to play video games.
[6] My cousin likes to practice yoga.
[7] My colleague really likes to attend concerts.

**Question:**
Oh, the weather has been so unpredictable lately, it was hot enough to wear short sleeves yesterday, but today I had to put on a jacket. Speaking of which, my favorite season is autumn, not too cold, not too hot, it's the most comfortable time for a walk. By the way, that coffee shop recommended by a friend last time seems pretty good, I should find some time to try it. What I wanted to ask is, what are the hobbies of the person who works in Shenzhen, Guangdong?

**Answer(Text):**
Sports.

**Choices:**
A. Traveling.
B. Photography.
C. Sports.
D. Reading.

**Answer(Choice):** C
**Answer(Retrieval):** [2, 3]
**Time:** April 4, 2024, Thursday 18:08

---

# H    Description of Human Evaluation Guidelines

## H.1    Guideline of Evaluation on User Profiles

Guideline: You will see some user profiles in the left column of the questionnaire. Please assess whether these user profiles are reasonable, and rate the rationality of them ranging from 1 to 5. Score 1 means the least reasonable, while score 5 means the most reasonable.

Here, reasonableness refers to: (1) Likely to exist in the real world, resembling a real user (realistic); (2) No inside conflicts or contradictions (consistent).

Here are some examples of unreasonable cases for reference:
(1) [1 point] The user's age is 24, but the related person is his grandson. (Logical error: A 24-year-old cannot have a grandson.)
(2) [2 points] The user's height is "(1) 175cm (2) 168cm (3) 172cm". (Generation error: Multiple values are given for a single attribute that can only have one value, like height.)
(3) [2-4 points] The user's phone number is 01234567891. (Unrealistic: The phone number does not seem real.)
(4) [2-4 points] The user's company name is Shanghai XX Company. (Unrealistic: The company name XX does not seem real.)

Tips: If there are no obvious unreasonable aspects, a score of 5 can be given; if there are serious errors, a score of 1-2 can be given; for other unrealistic elements, points can be deducted accordingly.

## H.2    Guideline of Evaluation on User Messages

Guideline: You will see some messages in the left column of the questionnaire. These messages are what the user said to the personal assistant while using it, i.e., the recorded user messages. Please assess the fluency, rationality, naturalness, and informativeness of these user messages, and score them ranging from 1 to 5.

**[Fluency]** The fluency of user messages refers to the correctness of the message text in terms of words, sentences, and grammar; whether the message text is coherent and conforms to language and expression habits, allowing for colloquial expressions. Score 1 means the least fluent, while score 5 means the most fluent.

Here are some examples that lack of fluency for reference:
(1) [1-2 point] Today day day day upwards to juggle night, I ate meat pork and or but rice. (Hardly understand due to lack of fluency.)
(2) [2-3 point] This night, I pork and rice, delicious. (Requires effort to guess due to lack of fluency, but can realize what it means.)

Tips: No obvious issues in fluency can be given a score of 5; serious errors can receive a score of 1 2; other elements affecting fluency can lead to a deduction of points as appropriate.

**[Rationality]** The rationality of the user message refers to: (1) it can be existed in the real world (2) without inside conflict and contradiction. Score 1 means the least rational, while score 5 means the most rational.

Here are some examples that lack of fluency for reference:
(1) [1 point] I am 24 years old, and my grandson is 2 years old. (It is impossible for a 24-year-old to have a grandson.)
(2) [2-3 point] Today is Monday, tomorrow is Wednesday. (Tomorrow cannot be Wednesday as the day after Monday is Tuesday.)

Tips: If there are no obvious unreasonable points, a score of 5 can be given; for serious errors, a score of 1-2 can be given; for other unreasonable elements, corresponding points can be deducted at discretion.

**[Naturalness]** The naturalness of a user message refers to whether the message closely resembles a real user message. Score 1 means the least natural, while score 5 means the most natural.

**[Informativeness]** The informativeness of user messages refers to whether these messages can provide rich and valuable information points. Information points are those points that can be queried about. Score 1 means the least informative, while score 5 means the most informative.

The following are some examples:
(1) [Low Informativeness] How is the weather today?
(2) [Medium Informativeness] How is the weather today? I plan to go to the park this afternoon.
(3) [High Informativeness] Today's weather is overcast turning to cloudy, it won't rain, I plan to go to the park this afternoon.

Highlight: You should have a general sense of the informativeness in the user's message during the pre-evaluation phase.

Additional Requirement: You should indicate the reason at the above critical points for deduction. If no major points for deduction exist, then there is no need to fill in this requirement.

### H.3   Guideline of Evaluation on Questions and Answers

Guideline: In the left column of the questionnaire, you will see (1) a list of user messages (2) a question (3) the textual answer (4) the multiple choices with the correct answer (5) the index list of retrieval targets. You should check the three aspects of the QAs, including the accuracy of textual answers, the accuracy of multiple-choice answers, and the accuracy of retrieval targets.

**[Accuracy of Textual Answers]** You need to check whether the textual answer is correct relative to the question based on the user's message list. If it is correct, please select the button [Correct], otherwise, please select the button [Incorrect].

**[Accuracy of Retrieval Targets]** Please judge the correctness of the retrieval targets in the Q&A. Retrieval targets refer to which messages (given in index form) from the user's message list are needed to obtain the textual answer to the question. Determine whether the retrieval targets are correct. If it is uniquely correct, please select the button [Correct], otherwise, please select the button [Incorrect].

Additional Requirement: You should indicate the reason for choosing [Incorrect]. If all of the above are correct, then there is no need to fill in this requirement.

