# OpenReview forum: "MemSim: A Bayesian Simulator for Evaluating Memory of LLM-based Personal Assistants"
_NeurIPS.cc/2025/Conference — NeurIPS 2025 poster_

### Official Review · Reviewer_k9DJ · 2025-06-12

**Clarity:** 2
**Significance:** 2
**Originality:** 2
**Rating:** 4
**Confidence:** 3

**Summary:**

This paper presents MemSim, an automated framework for evaluating the memory capabilities of large language model (LLM)-based personal assistants.  By constructing a Bayesian Relation Network (BRNet), the authors generate logically coherent and diverse user profiles.  These profiles serve as the foundation for a causal generation mechanism that simultaneously produces user messages and corresponding question-answer pairs from shared structured hints, mitigating hallucinations common in LLM-generated content.  Based on this approach, the authors build MemDaily, a comprehensive evaluation dataset that spans various memory task types, and introduce a benchmarking framework to evaluate the memory capability of LLM-based personal assistants.

**Questions:**

1.It would be helpful if the authors could elaborate on the motivation for choosing BRNet over other data modeling methods (e.g., knowledge graph-based sampling). The paper would be strengthened by either (1) providing theoretical or empirical evidence that BRNet yields higher diversity or consistency in user profiles, or (2) including a comparative experiment demonstrating that datasets generated with BRNet outperform those generated by alternative methods in terms of QA reliability and coverage.

2.Although the authors designed multiple internal baselines to compare dataset attributes, all of these baselines rely solely on their self-constructed simulation environment.  The lack of horizontal comparison with existing benchmarks—such as the EpisodicQA-generated datasets [1]—limits the persuasiveness of the proposed method.  [1] Huet A, Houidi Z B, Rossi D. Episodic Memories Generation and Evaluation Benchmark for Large Language Models. arXiv preprint arXiv:2501.13121, 2025.

3.It may be beneficial to include datasets that are more reflective of real-world scenarios—such as multi-turn dialogue settings (e.g., a user says “I’m going to see the dentist” today, and three days later mentions “The doctor reminded me to check for cavities”)—to better evaluate whether the method can maintain and retrieve information consistently in context.

4.The current benchmark evaluation (Section 5.2) is conducted solely using the GLM-4-9B model, which may limit the generalizability of the findings. It is recommended to include evaluations using multiple LLMs (e.g., LLaMA) to provide a more comprehensive assessment of the dataset’s robustness and applicability.

**Ethical Concerns:**

["NO or VERY MINOR ethics concerns only"]

**Final Justification:**

I have read the authors' response. Thanks for the rebuttal. The response addressed some of my concerns. I will raise my rating accordingly. Considering the overall quality (novelty, experiments, method, idea), I revised my rating score as 4.

**Limitations:**

The evaluation protocol relies heavily on subjective human ratings and lacks a standardized, reproducible scoring framework, which may undermine the consistency and generalizability of the results.

**Quality:**

3

**Strengths And Weaknesses:**

Strengths:

1.The introduction of the Bayesian Relation Network (BRNet) to simulate user profiles and model causal dependencies among attributes through a graph structure enhances the diversity and consistency of user information.

2.The MemDaily dataset constructed based on MemSim covers six common types of memory tasks encountered in real-world scenarios, providing a unified benchmark for subsequent related research.

Weaknesses:

1.While the paper presents a systematic approach to constructing a factual QA dataset, the evaluation scope is limited to explicit and structured factual memory.  It overlooks higher-level memory types, such as dynamically evolving memory representations from long-term interactions, as well as implicit forms of memory like user preferences, intentions, and emotional tendencies.

2.The authors attempt to mitigate hallucination issues in LLMs through a causality-based generation mechanism guided by structured hints.   However, the generation of user messages and questions still heavily relies on the language modeling capabilities of LLMs, and no real user dialogues or log data are used for validation.   As a result, the evaluation may not accurately reflect the memory capabilities of LLMs in open-domain settings.

3.The paper relies heavily on human subjective ratings, which are treated as the primary evaluation metric.   This approach is somewhat unconvincing due to its strong subjectivity and limited reproducibility.   A more robust evaluation strategy would prioritize structural consistency or other quantitative metrics, using human judgment as a supplementary measure.   Besides, the paper should clearly specify the human evaluation process, including the criteria, procedures, and consistency scores

---

> ### Author Rebuttal · Authors · 2025-07-29
>
> Dear Reviewer k9DJ,
>
> Thanks so much for your precious time in reading and reviewing our paper. However, I believe there may be some misunderstandings about our paper, and I hope the following rebuttal can make a clarification and change your perspective on our work. In the following, we try to alleviate your concerns one by one.
>
>
> **For Weakness 1: "While the paper presents a systematic approach ... tendencies."**
>
> **Response:**
>
> Thank you very much for your valuable comments. In this paper, we indeed primarily focus on users' factual information rather than implicit preferences. However, our proposed MemSim framework can be readily extended to accommodate higher-level memory types. For instance, in user preference scenarios, we can similarly construct graph structures based on BRNet to represent relationships between user preferences, and leverage Causal Generation Mechanisms to generate question-answer pairs based on users' implicit preferences. Meanwhile, we can utilize LLMs to rewrite implicit preferences into concrete factual information, thereby making high-level preferences explicit and enabling the evaluation of higher-level memory.
>
> Regarding evolving memory representations from long-term interactions, we have implemented a model-agnostic dynamic comprehensive evaluation of memory, where the corresponding methods need to complete the processes of storage, retrieval, and utilization. Specifically, in the anonymous code repository referenced at the end of the abstract, we have constructed a complete temporal data flow evaluation where user messages are provided sequentially with timestamps, as demonstrated in `benchmark/Timeflow.py`. In future work, we will further refine the generation pipeline for higher-level memory types.
>
>
> **For Weakness 2: "The authors attempt to mitigate hallucination ... open-domain settings."**
>
> **Response:**
>
> Thank you for your valuable comments. We agree with your point that hallucinations arising from LLM generation capabilities cannot be completely avoided. However, our proposed method can mitigate the degree of hallucination during the data generation process, as difficult problems are more prone to hallucinations, and our approach can decompose the challenging generation process into simpler multi-step procedures through causal mechanisms.
>
> Furthermore, we conducted evaluation experiments on the generated data in Section 4. Through these experiments, we found that the reliability of our method shows significant improvement, and the generated question-answer pairs also demonstrate high accuracy. Additionally, other existing papers [2] also rely on LLM-based generation and cannot completely eliminate LLM hallucinations, but can only mitigate them as much as possible.
>
>
> **For Weakness 3: "The paper relies heavily on human subjective ratings ... consistency scores."**
>
> **Response:**
>
> Thank you for your comments. Regarding dataset generation evaluation, many works use human verification methods to assess data quality, such as [3]. We use both human annotators and LLM-based evaluations for objective scoring.
>
> Here are the detailed human evaluation steps. We recruited six well-educated human experts to score multiple aspects of our Memdaily dataset. We designed a standard evaluation pipeline with clear instructions, fair scoring, and reasonable comparisons. Our evaluations focus on user profiles, user messages, and QAs (mentioned in Section 4).
>
> Our pipeline includes five steps: (1) Human evaluator recruitment (2) Guideline and questionnaire design (3) Web page construction and deployment (4) Pre-evaluation (5) Formal evaluation.
>
> We recruited six human experts as evaluators, all holding at least bachelor's degrees to ensure correct understanding of evaluation questions and reasonable feedback. We designed detailed guidelines instructing evaluators on the evaluation process. The guideline includes three parts corresponding to the three evaluation aspects in Section 4.
>
> Due to response length limitations, we will provide a description of the human evaluation. The guideline clearly defines score meanings and ensures consistent scoring standards across evaluators. For consistency scores, see means and standard deviations in Tables 3 and 4.
>
>
> **For Questions 1 : "It would be helpful if the authors could elaborate on the motivation ... coverage."**
>
> **Response:**
>
> Thank you for your question. Regarding knowledge graph-based sampling, in our paper, we just need conditional probability distributions among variables, and utilize ancestral sampling to obtain different user profiles, instead of calculating the complex and high-dimensional joint probability distribution. These conditional probability distributions should be introduced into BRNet as prior knowledge, before sampling user profiles. For BRNet, we also provide comparisons of the user profiles it generates in Table 3. As shown in Table 3, MemSim outperforms other baselines on Rationality, demonstrating the effectiveness of BRNet as an ablation study.
>
> Regarding QA reliability and coverage, we have included the evaluation results in Appendix B of our paper. We sample approximately 20% of all the trajectories in MemDaily and employ human evaluators to verify the correctness of their ground truths. Specifically, the evaluators are required to examine the ground truths and report their accuracy in Table 7. We find that MemDaily significantly ensures the accuracy of the answers provided for constructed questions.
>
> **For Questions 2: "Although the authors designed multiple ... EpisodicQA-generated datasets ..."**
>
> **Response:**
>
> Thank you for your comments. We will add comparative content with the following two papers in the revised version. MemSim proposes a complete Bayesian simulator specifically designed to automatically construct reliable question-answer pairs for evaluating LLM memory capabilities, while ensuring diversity and scalability. In contrast, the Episodic Memories Generation research is still in the conceptual exploration stage, primarily focusing on how to integrate episodic memory capabilities into LLMs, but lacks a mature automated evaluation system. MemSim provides an operational technical solution rather than just a theoretical framework by introducing Bayesian Relation Networks (BRNet) and causal generation mechanisms.
>
>
> **For Questions 3: "It may be beneficial to include datasets that are more reflective of real-world scenarios ... consistently in context."**
>
> **Response:**
>
> Thank you for your suggestion. In fact, our method can also be applied to extract information from real-world data. You can extract entities and relationships from existing datasets using methods such as LLMs, construct BRNet according to the method described in Section 3.2 of our paper, and then perform sampling and generation on the BRNet to construct corresponding user messages, questions, and answers. We have also applied MemSim to real business scenarios and achieved promising results. In future work, we plan to supplement relevant datasets that are abstracted and extracted from real-world data.
>
> **For Questions 4: "The current benchmark ... solely using the GLM-4-9B model ... applicability."**
>
> **Response:**
>
> Thank you for your suggestion. To better compare the performance across different models, we have additionally included evaluation results for Qwen2.5. The experimental results are as follows.
>
> Qwen2.5-7B (Accuracy):
>
> | Methods   | Simp.       | Cond.       | Comp.       | Aggr.       | Post.       | Noisy       |
> | --------- | ----------- | ----------- | ----------- | ----------- | ----------- | ----------- |
> | FullMem   | 0.980±0.008 | 0.974±0.017 | 0.876±0.029 | 0.312±0.045 | 0.853±0.003 | 0.966±0.006 |
> | RetrMem   | 0.880±0.005 | 0.878±0.010 | 0.709±0.019 | 0.229±0.027 | 0.787±0.024 | 0.799±0.008 |
> | ReceMem   | 0.809±0.023 | 0.797±0.010 | 0.512±0.026 | 0.145±0.012 | 0.739±0.033 | 0.743±0.046 |
> | NonMem    | 0.526±0.017 | 0.424±0.027 | 0.183±0.005 | 0.255±0.013 | 0.629±0.033 | 0.376±0.010 |
> | NoisyMem  | 0.512±0.017 | 0.424±0.050 | 0.262±0.017 | 0.301±0.017 | 0.637±0.032 | 0.373±0.020 |
> | OracleMem | 0.982±0.005 | 0.976±0.005 | 0.909±0.023 | 0.450±0.089 | 0.880±0.020 | 0.978±0.008 |
>
> Qwen2.5-7B (Recall):
>
> | Methods   | Simp.       | Cond.       | Comp.       | Aggr.       | Post.       | Noisy       |
> | --------- | ----------- | ----------- | ----------- | ----------- | ----------- | ----------- |
> | FullMem   | 0.939±0.006 | 0.884±0.007 | 0.918±0.009 | 0.601±0.016 | 0.863±0.003 | 0.872±0.019 |
> | RetrMem   | 0.726±0.013 | 0.717±0.023 | 0.846±0.008 | 0.516±0.029 | 0.691±0.014 | 0.649±0.014 |
> | ReceMem   | 0.596±0.001 | 0.524±0.025 | 0.700±0.010 | 0.223±0.012 | 0.501±0.023 | 0.503±0.031 |
> | NonMem    | 0.319±0.013 | 0.391±0.019 | 0.414±0.048 | 0.183±0.008 | 0.319±0.006 | 0.562±0.017 |
> | NoisyMem  | 0.575±0.025 | 0.525±0.014 | 0.725±0.012 | 0.454±0.007 | 0.487±0.025 | 0.660±0.019 |
> | OracleMem | 0.406±0.021 | 0.609±0.035 | 0.782±0.016 | 0.562±0.025 | 0.605±0.009 | 0.585±0.023 |
>
> Based on the experimental results for Qwen2.5-7B, a clear performance hierarchy emerges across different memory mechanisms, with FullMem and OracleMem consistently achieving the highest accuracy scores. This is broadly consistent with the conclusions drawn from GLM-4-9B.
>
> **We sincerely thank you for your time to review our paper and comments on it. I hope the rebuttal can make a clarification of the misunderstandings and change your perspective on our work. If you have further questions, we are very happy to discuss them.**
>
> **References:**
>
> [1] Huang, Lei, et al. "A survey on hallucination in large language models: Principles, taxonomy, challenges, and open questions." *ACM TOIS* 43.2 (2025): 1-55.
>
> [2] Wu, Di, et al. "Longmemeval: Benchmarking chat assistants on long-term interactive memory." *arXiv:2410.10813* (2024).
>
> [3] Maharana, Adyasha, et al. "Evaluating very long-term conversational memory of llm agents." *arXiv:2402.17753* (2024).

---

> > ### Author Response · Authors · 2025-08-06
> >
> > Dear Reviewer k9DJ,
> >
> > Thank you again for your detailed comments, which we believe are very important in improving our paper.
> >
> > We have tried our best to address the concerns one by one. As the discussion deadline approaches, we eagerly await your feedback on our responses.
> >
> > If you have further questions, we are very happy to discuss them. We really hope our efforts can alleviate your concerns.
> >
> > Sincerely,
> >
> > Submission 4675 Authors

---

> ### Comment · Area_Chair_qxFC · 2025-08-09
>
> Dear Reviewer k9DJ,
>
> Thank you for your time and effort in reviewing the submissions and for providing valuable feedback to the authors.
>
> It is much appreciated that you could acknowledge the rebuttal by clicking the “Rebuttal Acknowledgement” button at your earliest convenience, and engage the discussion. In particular, if your evaluation of the paper has changed, please update your review and explain the revision.
>
> This step ensures smooth communication and helps us move forward efficiently with the review process.
>
> We sincerely appreciate your dedication and collaboration.
>
> Best regards, AC

---

### Official Review · Reviewer_rBqV · 2025-06-29

**Clarity:** 3
**Significance:** 2
**Originality:** 3
**Rating:** 4
**Confidence:** 4

**Summary:**

This paper introduces MemSim, a Bayesian simulator designed to automatically generate reliable datasets for evaluating the memory of LLM-based personal assistants, thereby addressing the high cost of human annotation and the hallucination issues in existing automatic methods. MemSim leverages a Bayesian Relation Network to model user profiles with causal dependencies and a novel generation mechanism to create user messages and question-answer pairs from shared, structured "hints." Using this simulator, the authors created the MemDaily dataset to test various memory mechanisms, revealing that current LLM assistants struggle significantly with complex comparative and aggregative questions, especially when faced with noisy or irrelevant information.

**Questions:**

1: Your current evaluation treats memory as static fact retrieval, but real personal assistants need to handle evolving user information, conflicting updates, and temporal dependencies. How would you extend MemDaily to evaluate these dynamic aspects of memory management?

2: You claim the approach is scalable, but BRNet requires manual specification of entities, attributes, and causal relationships for each domain. Can you provide concrete evidence of how easily this transfers to new domains (e.g., professional contexts, healthcare scenarios)? What would be required to adapt MemSim to evaluate memory in coding assistants or research agents?

3: While you position this as an evaluation methodology paper, the core technical contributions appear to be primarily about reliable dataset generation with BRnet sampling and causal strurcual prompting method rather than novel evaluation metrics or frameworks. What makes this fundamentally different from a benchmark paper that happens to introduce a novel dataset generation method?

**Ethical Concerns:**

["NO or VERY MINOR ethics concerns only"]

**Final Justification:**

reviewers has addressed most of my concern; didn't give a higher score since I think i was looking for more connections with other agentic components

**Limitations:**

yes.

**Paper Formatting Concerns:**

No.

**Quality:**

3

**Strengths And Weaknesses:**

Strengths:
1: The paper addresses a genuine and important problem in LLM agent evaluation. The technical approach is well-motivated and the experimental evaluation is comprehensive, including both dataset quality assessment and memory mechanism benchmarking.

2: The combination of Bayesian networks for user profile generation and causal generation mechanisms is novel for this application domain.

Weakness:
1: While the authors claim scalability, the approach requires manual design of entity-attribute relationships and causal structures for each domain, which may limit generalizability.

2: Disconnect from Recent Advances: The paper doesn't connect with recent developments in agentic systems like tool use, reflection mechanisms, or planning capabilities that are increasingly important for personal assistants with memories. (e.g. only tests retrieval and basic reasoning, missing evaluation of how memory integrates with other agent capabilities like tool calling or multi-step planning). Also as for memory systems, modern memory systems often incorporate semantic similarity, knowledge graphs, or learned embeddings. The lack of such comparison limits the scope of this paper and I am skeptical about its value to the community given the rapid advancement in modern AI Agents.

---

> ### Author Rebuttal · Authors · 2025-07-29
>
> Dear reviewer rBqV,
>
> Thanks so much for your precious time in reading and reviewing our paper, and we are encouraged by your positive feedback. In the following, we try to alleviate your concerns one by one:
>
>
>
> **For Weakness 1: "While the authors claim scalability, the approach requires manual design of entity-attribute relationships and causal structures for each domain, which may limit generalizability."**
>
> **Response:**
>
> Thank you for your valuable question. While BRNet does require entity-attribute relationships and causal structures, this process can be automated by leveraging LLMs to extract the corresponding entities, attributes, and features, thereby enabling automatic construction of the required datasets. For example, with novel data, we can extract character roles, attributes, and relationships to construct the BRNet, derive questions and answers based on these relationship chains, and generate statements as memory observations for the agent.
>
> In our paper, we focus on designing the generative paradigm of BRNet. The construction of relationships required by BRNet from raw information can be achieved through LLM-based extraction or direct rule-based summarization. The rule-based construction method is suitable for scenarios with clear rules and simple relationships, producing data with lower noise that is more robust. The LLM-based automatic extraction of relationships and entities is applicable to large-scale scenarios or cases where explicit rules cannot be designed, though it may introduce noise during the LLM extraction process, requiring additional human correction of the resulting BRNet.
>
> We believe this flexibility in construction approaches actually enhances the scalability of our method, as it can adapt to different domain requirements and data availability constraints.
>
>
>
> **For Weakness 2: "Disconnect from Recent Advances ... rapid advancement in modern AI Agents."**
>
> **Response:**
>
> Thank you for your insightful question. We strongly agree that agents require not only memory but also additional mechanisms such as tool use, reflection mechanisms, and planning capabilities to assist in memory implementation. However, the evaluation dataset construction method proposed in this paper focuses more on a model-agnostic setting, meaning that any memory system can be evaluated on the dataset constructed by our method. The research on how other mechanisms affect memory that you mentioned can also be implemented on our dataset by simply fixing the memory method and varying different additional capabilities to compare their performance. For example, to study the impact of reasoning capabilities on memory, one could fix the memory method as RAG-based long-term memory and then compare reasoning structures such as CoT, ToT, and GoT. We would be very interested to see future work explore the influence of other agent capabilities on memory.
>
> Furthermore, regarding the memory technologies you mentioned, such as semantic similarity, knowledge graphs, and learned embeddings, these are all components within memory systems that still require the memory framework of storage, retrieval, and utilization to function [1, 2]. Our model-agnostic setting can precisely accommodate these memory components you mentioned by fixing other parts and then modifying the component under study to compare performance. In the future, we will also optimize and improve the evaluation of the integration between other agent capabilities and memory.
>
> We believe our framework provides a solid foundation for systematic evaluation that can evolve with the rapid advances in agentic systems, and we welcome the community to build upon this work to explore these important research directions.
>
>
>
> **For Question 1: "Your current evaluation treats memory as static fact retrieval ... dynamic aspects of memory management?"**
>
> **Response:**
>
> Thank you very much for your insightful question. In fact, our proposed evaluation method is model-agnostic for memory systems and provides a comprehensive evaluation of memory systems as a whole. The corresponding methods need to complete the processes of storage, retrieval, and utilization, rather than simple static retrieval. Specifically, in the anonymous code repository referenced at the end of the abstract, we have constructed a complete temporal data flow evaluation where user messages are given sequentially with timestamps, as shown in `benchmark/Timeflow.py`. Therefore, our evaluation has already implemented dynamic extension with temporal information.
>
> Regarding evolving user information and conflicting updates, these can be addressed through the question generation process of the Causal Generation Mechanism by inserting different timestamps for conflicting information, which can seamlessly integrate with `Timeflow.py`. This design allows our framework to naturally handle the dynamic aspects of memory management that you mentioned, including temporal dependencies and information conflicts that occur over time.
>
> We believe this temporal evaluation capability demonstrates that our approach goes beyond static fact retrieval and can effectively assess how memory systems handle the complexities of real-world personal assistant scenarios.
>
>
>
> **For Question 2: "You claim the approach is scalable, but BRNet requires manual specification ... or research agents?"**
>
> **Response:**
>
> Thank you for your question. For BRNet, construction can be achieved through both rule-based approaches and LLM-based extraction of entities and relationships for autonomous construction. The rule-based construction method is suitable for scenarios with clear rules and simple relationships, producing data with lower noise that is more robust. The LLM-based automatic extraction of relationships and entities is applicable to large-scale scenarios or cases where explicit rules cannot be designed, though it may introduce noise during the LLM extraction process, requiring additional human correction of the resulting BRNet. As you mentioned, if the attribute set is very large and causal relationships are complex, rule-based approaches would be very challenging, so adopting automatic entity and relationship construction is the appropriate choice.
>
> For example, in financial and healthcare domains, we can use LLMs to extract key entities, attributes, and corresponding relationships based on large-scale domain expert knowledge, and integrate specific application scenarios (such as Alice's situation) to construct BRNet. Through sampling, we can obtain corresponding questions and answers, generate corresponding statements (user messages), and produce the final evaluation dataset.
>
> For coding assistants, we can consider code files as the raw dataset, extract different classes as entities, extract different methods as attributes, and derive relationships between various classes and attributes to construct the corresponding memory evaluation dataset. For research agents, we can use knowledge from a specific domain as raw data, extract nouns as entities, their features as attributes, and extract relationships between them, applying MemSim to obtain the final evaluation dataset.
>
> This flexibility in construction approaches demonstrates the scalability of our method across diverse domains, from personal scenarios to professional contexts, by adapting the entity-attribute-relationship extraction process to domain-specific characteristics while maintaining the same underlying evaluation framework.
>
>
>
> **For Question 3: "While the authors claim scalability, the approach requires manual design of entity-attribute relationships and causal structures for each domain, which may limit generalizability."**
>
> **Response:**
>
> Thank you for your question. I agree with your observation that our core contribution is proposing a data generation method based on BRNet and the Causal Generation Mechanism. However, we focus on designing memory data generation methods from an abstract methodological perspective, rather than concentrating on a specific dataset (such as the Daily Life scenario) and its evaluation, although it can be conveniently transferred and applied to generate new datasets. As we mentioned above, our method can also be applied to other domains, such as finance and healthcare. Additionally, we analyze and explore the current performance of explicit memory systems, discussing the advantages and disadvantages of different models. Therefore, we believe this paper differs from a typical benchmark paper.
>
> Our work provides a systematic framework for generating evaluation data for memory systems across domains, rather than simply presenting a fixed benchmark. The methodology we propose enables researchers to create domain-specific memory evaluation datasets using the same principled approach, which we believe constitutes a valuable methodological contribution to the field of memory evaluation for AI agents.
>
>
>
> **We sincerely thank you for your time to review our paper, and we also thanks for your insightful comments, which, we believe, are very important to improve our paper. We hope our responses can address your concerns. If you have further questions, we are very happy to discuss them.**
>
>
>
> **References:**
>
> [1] Zhang, Zeyu, et al. "A survey on the memory mechanism of large language model based agents." *ACM Transactions on Information Systems* (2024).
>
> [2] Zhang, Zeyu, et al. "MemEngine: A Unified and Modular Library for Developing Advanced Memory of LLM-based Agents." *Companion Proceedings of the ACM on Web Conference 2025*. 2025.

---

> > ### Author Response · Authors · 2025-08-06
> >
> > Dear Reviewer rBqV,
> >
> > Thank you again for your detailed comments, which we believe are very important in improving our paper.
> >
> > We have tried our best to address the concerns one by one. As the discussion deadline approaches, we eagerly await your feedback on our responses.
> >
> > If you have further questions, we are very happy to discuss them. We really hope our efforts can alleviate your concerns.
> >
> > Sincerely,
> >
> > Submission 4675 Authors

---

> > ### Comment · Reviewer_rBqV · 2025-08-06
> > **reply**
> >
> > Thanks for your detailed rebuttal, which indeed addressed most of my concerns. Personally, I would still love to see how such evaluation methods interplay with other agentic components. I believe this is important because most LLM agent research and memory design is fairly empirically driven, and there is no universal rule on what memory approach you should adopt. It depends on your overall agent design as well as the specific tasks. While it's nice that the method leverages a Bayesian Network as theoretical grounding, ultimately, to benefit the research community, a more comprehensive design covering those aspects would be very appreciated. However, this is my subjective opinion and I understand a 9-page paper can't cover everything, so I will leave the AC to make this judgment call. I will raise my score to 4.

---

> > > ### Author Response · Authors · 2025-08-06
> > >
> > > Dear Reviewer rBqV,
> > >
> > > Thank you for your thoughtful feedback. We are greatly encouraged by your recognition of our paper.
> > >
> > > The point about the interplay between evaluation methods and other agentic components is very insightful, which is worthwhile to explore further (such as reasoning and planning). Additionally, we agree that a more comprehensive framework considering overall agent design would be valuable for evaluation,and we will continue working toward this goal. Although space constraints limit the scope of this paper, we are committed to expanding this work to explore these important points in future research.
> > >
> > > We are grateful for your constructive suggestions, which will help guide our continued efforts to advance this area.
> > >
> > > Sincerely,
> > >
> > > Submission 4675 Authors

---

> > > ### Author Response · Authors · 2025-08-07
> > >
> > > Dear Reviewer rBqV,
> > >
> > > Thank you very much for indicating that you would like to raise our paper's score to 4. We greatly appreciate your reconsideration and support.
> > >
> > > We noticed that the score may not have been updated in the review system yet. Would it be possible for you to kindly update the score in the system when convenient? This would be very helpful for our submission process.
> > >
> > > Thank you again for your time and constructive feedback.
> > >
> > > Sincerely,
> > >
> > > Submission 4675 Authors

---

### Official Review · Reviewer_GeDE · 2025-07-03

**Clarity:** 2
**Significance:** 3
**Originality:** 3
**Rating:** 4
**Confidence:** 4

**Summary:**

MemSim is a Bayesian simulator that automatically generate reliable, diverse, and scalable evaluation datasets MemDaily.  MemDaily assesses the memory capabilities of LLM-based personal assistants. MemSim uses BRNet, which models hierarchical user profiles by sampling entities and their attributes in a causally structured, loop-free graph. A causal generation mechanism uses these profiles to produce user messages and construct QA pairs to reduce hallucination. MemDaily comprises six QA types, reflecting real-world query complexity. Extensive human and automated evaluations demonstrate that MemDaily maintains high accuracy, fluency, and diversity. The authors benchmark various memory mechanisms on this dataset, including full memory, recent memory, and retrieved memory, measuring both effectiveness and efficiency.

**Questions:**

see Weaknesses

**Ethical Concerns:**

["NO or VERY MINOR ethics concerns only"]

**Final Justification:**

My preference is positive and main concern is about the evaluation, such as the generated questions. During rebuttal, the authors have resovled most of my concerns. So, I keep the positive view.

**Quality:**

2

**Strengths And Weaknesses:**

Strengths:
1.	MemSim’s pipeline is automatic, which saves expensive human annotation. It generates QAs from simulated user profiles, which enables large-scale and reproducible memory evaluation.
2.	The proposed method derives user messages and QAs from the same hints sampled from BRNet. This method effectively reduces hallucination as shown in Table 5.
3.	The authors propose a comprehensive QA taxonomy, including the 6 QA types (simple, conditional, comparative, aggregative, post-processing, and noisy). This taxonomy covers a wide range of reasoning scenarios, which provides a better and more detailed evaluation guideline for memory mechanisms.
4.	MemDaily dataset has a high diversity in the simulated profiles and messages, as proven by its Shannon–Wiener score. It shows that BRNet’s hierarchical sampling and the causal mechanism are useful in generating varied user profiles.
5.	This papers benchmarks the memory mechanisms in an end-to-end manner, including evaluateing multiple memory strategies on both vanilla and noisy variants of the dataset. The authors conduct extensive experiments and provide clear metrics for effectiveness and efficiency.


Weaknesses:
1.	The experiments uses MemDaily to evaluate only GLM-4-9B. It would be better to evaluate models from different families or models of different sizes. A comparison of the results from different models can better showcase the benchmark’s ability to evaluate models.
2.	Although you mention in check point 14 (Line824 to 838) that you have included full text instructions given to participants, I could not  find any from the paper, including appendix. This brings that problem that your human evaluation to user profile (line236 - 237) and user messages (Line 254 to 255) become very unclear. Human evaluators give scores scale from 1 – 5, but what is a score 1 profile like? What is a score 5 profile like? What aspects should the evaluator look into a profile? Would evaluator A’s score 3 be the same as evaluator B’s score 3? Without the guideline, your human evaluations are very weak.
3.	Some generated questions are still very artificial, especially the multi-hop questions and post-processing questions. For instance, in appendix F3 (Line 534 - 535), “where does the person with only a high school education work now?” or (line 540-541),” Which of the following descriptions matches the work location of the person whose birthday is July 15th?” I hardly think a question like this will appear from any real user. There is still a gap in the distribution of actual encountered questions and the generated ones.
4.	Your method of controlling dataset quality is through adding more noise from social media. However, since your useful information are model generated, they naturally have a different distribution from the real message from social media. And hence the negative samples inserted are not high-quality negatives. Hence the difference of performance between different difficulties are not very obvious (table 11 – table 22)
5.	There are some further areas of improvement.
Like a) including a memory editing mechanism in the benchmark through dynamic dialogue
And b) your college and relative entities are a little toyed. (line 518-522, table 23). In real life a person’s relationship is far beyond colleagues and relatives. It would be better to set up the colleges/relatives also as individual entities, and each entity can be a (self), the user entity can be connected to many other entities with more complex relationships.

---

> ### Author Rebuttal · Authors · 2025-07-29
>
> Dear reviewer GeDE,
>
> Thanks so much for your precious time in reading and reviewing our paper, and we are encouraged by your positive feedback. In the following, we try to alleviate your concerns one by one:
>
>
>
> **For Weakness 1: "The experiments uses MemDaily to evaluate only GLM-4-9B... ability to evaluate models."**
>
> **Response:**
>
> Thank you for your suggestion. To better compare the performance across different models, we have additionally included evaluation results for Qwen2.5. The experimental results are as follows.
>
> Qwen2.5-7B (Accuracy):
>
> | Methods   | Simp.       | Cond.       | Comp.       | Aggr.       | Post.       | Noisy       |
> | --------- | ----------- | ----------- | ----------- | ----------- | ----------- | ----------- |
> | FullMem   | 0.980±0.008 | 0.974±0.017 | 0.876±0.029 | 0.312±0.045 | 0.853±0.003 | 0.966±0.006 |
> | RetrMem   | 0.880±0.005 | 0.878±0.010 | 0.709±0.019 | 0.229±0.027 | 0.787±0.024 | 0.799±0.008 |
> | ReceMem   | 0.809±0.023 | 0.797±0.010 | 0.512±0.026 | 0.145±0.012 | 0.739±0.033 | 0.743±0.046 |
> | NonMem    | 0.526±0.017 | 0.424±0.027 | 0.183±0.005 | 0.255±0.013 | 0.629±0.033 | 0.376±0.010 |
> | NoisyMem  | 0.512±0.017 | 0.424±0.050 | 0.262±0.017 | 0.301±0.017 | 0.637±0.032 | 0.373±0.020 |
> | OracleMem | 0.982±0.005 | 0.976±0.005 | 0.909±0.023 | 0.450±0.089 | 0.880±0.020 | 0.978±0.008 |
>
> Qwen2.5-7B (Recall):
>
> | Methods   | Simp.       | Cond.       | Comp.       | Aggr.       | Post.       | Noisy       |
> | --------- | ----------- | ----------- | ----------- | ----------- | ----------- | ----------- |
> | FullMem   | 0.939±0.006 | 0.884±0.007 | 0.918±0.009 | 0.601±0.016 | 0.863±0.003 | 0.872±0.019 |
> | RetrMem   | 0.726±0.013 | 0.717±0.023 | 0.846±0.008 | 0.516±0.029 | 0.691±0.014 | 0.649±0.014 |
> | ReceMem   | 0.596±0.001 | 0.524±0.025 | 0.700±0.010 | 0.223±0.012 | 0.501±0.023 | 0.503±0.031 |
> | NonMem    | 0.319±0.013 | 0.391±0.019 | 0.414±0.048 | 0.183±0.008 | 0.319±0.006 | 0.562±0.017 |
> | NoisyMem  | 0.575±0.025 | 0.525±0.014 | 0.725±0.012 | 0.454±0.007 | 0.487±0.025 | 0.660±0.019 |
> | OracleMem | 0.406±0.021 | 0.609±0.035 | 0.782±0.016 | 0.562±0.025 | 0.605±0.009 | 0.585±0.023 |
>
> Based on the experimental results for Qwen2.5-7B, a clear performance hierarchy emerges across different memory mechanisms, with FullMem and OracleMem consistently achieving the highest accuracy scores, followed by moderate performance from RetrMem and ReceMem, while NonMem and NoisyMem demonstrate substantially degraded performance (0.183-0.637 range). The results reveal task-specific challenges, particularly evident in the Aggregation task where all methods show reduced accuracy, and demonstrate a trade-off between accuracy and recall metrics, with some methods like OracleMem showing lower recall despite high accuracy, suggesting potential overfitting or conservative prediction behavior. This is broadly consistent with the conclusions drawn from GLM-4-9B.
>
>
> **For Weakness 2: "Although you mention in check point 14 (Line824 to 838) that you have included full text instruction ... are very weak."**
>
> **Response:**
>
> Thank you very much for your valuable feedback. You are absolutely right, and we apologize for this omission. Below is the description of our human evaluation guidelines, which clearly defines what each score represents and ensures consistent scoring standards across different evaluators. We will include this description in the revised version of the paper.
>
> **Guideline of Evaluation on User Profiles.**
>
> *Guideline: You will see some user profiles in the left column of the questionnaire. Please assess whether these user profiles are reasonable, and rate the rationality of them ranging from 1 to 5. Score 1 means the least reasonable, while score 5 means the most reasonable. Here, reasonableness refers to: (1) Likely to exist in the real world, resembling a real user (realistic); (2) No inside conflicts or contradictions (consistent).*
>
> *Here are some examples of unreasonable cases for reference:*
>
> *(1) [1 point] The user's age is 24, but the related person is his grandson. (Logical error: A 24-year-old cannot have a grandson.)*
>
> *(2) [2 points] The user's height is "(1) 175cm (2) 168cm (3) 172cm". (Generation error: Multiple values are given for a single attribute that can only have one value, like height.)*
>
> *(3) [24 points] The user's phone number is 01234567891. (Unrealistic: The phone number does not seem real.)*
>
> ......
>
> *Tips: If there are no obvious unreasonable aspects, a score of 5 can be given; if there are serious errors, a score of 12 can be given; for other unrealistic elements, points can be deducted accordingly.*
>
> **Guideline of Evaluation on User Messages.**
>
> *Guideline: You will see some messages in the left column of the questionnaire. These messages are what the user said to the personal assistant while using it, i.e., the recorded user messages. Please assess the fluency, rationality, naturalness, and informativeness of these user messages, and score them ranging from 1 to 5.*
>
> *[Fluency] The fluency of user messages refers to the correctness of the message text in terms of words, sentences, and grammar*
>
> ...
>
> *Additional Requirement: You should indicate the reason at the above critical points for deduction. If no major points for deduction exist, then there is no need to fill in this requirement.*
>
> **For Weakness 3: "Some generated questions are still very artificial... gap in the distribution of actual encountered questions and the generated ones"**
>
> **Response:**
>
> Thank you very much for your valuable feedback. We acknowledge the gap between generated data and real-world scenarios, which stems from two main factors. First, LLM-synthesized data sources may create distributional bias compared to real-world scenarios. A solution is generating data based on real datasets by extracting entities and relationships from actual datasets, constructing a BRNet using our Section 3.2 methodology, then performing sampling and generation to create more reality-grounded user messages, questions, and answers. Second, rule-based generation processes may also cause distributional bias. We could automatically extract relationships and entities from source data using LLMs and construct corresponding Causal Generation Mechanisms for subsequent generation. Rule-based construction suits scenarios with clear rules and simple relationships, producing lower-noise data. LLM-based automatic extraction works better for large-scale scenarios or cases where explicit rules cannot be easily designed, though it may introduce noise during extraction, requiring human verification to correct the resulting BRNet. We will further investigate methods for generating data based on real-world scenarios to better bridge the gap between synthetic and authentic question distributions.
>
> **For Weakness 4: "Your method of controlling dataset quality is through adding more noise ... difficulties are not very obvious (table 11 – table 22) ."**
>
> **Response:**
>
> Thank you for your valuable comment. You are absolutely right that the noise we inject does not have exactly the same distribution as the original user messages, but we believe it has reasonable justification for the following reasons:
>
> 1. **Similar semantic distribution.** In our Daily Life scenario, user messages primarily focus on human daily activities, while the noise we introduce comes from news trending topics, which also belong to human daily life. The content expressed by both sources is generally similar in nature.
> 2. **Simulation of multi-source data in real scenarios.** In real-world scenarios, an agent's memory typically includes not only user-provided content but also additional information sources, such as important information from other applications. These information sources may not have exactly the same distribution as user-provided content, but they remain important. Furthermore, long-term information storage and retrieval is a key focus of current memory mechanisms and should be our primary emphasis.
> 3. **Scalability and efficiency considerations.** Constructing noise with perfectly consistent distribution would require generation based on the original profiles, carefully avoiding potential conflicts between user messages. This is difficult to achieve at scale and lacks reliability. Using out-of-distribution information avoids this issue and is more efficient.
> 4. **Consistency with existing work.** Most existing work adopts similar approaches for noise injection. For example, [1] also uses long documents to construct needle-in-a-haystack tasks.
>
> We acknowledge that this approach has limitations, but we believe it provides a reasonable balance between realism, scalability, and evaluation effectiveness for assessing long-term memory capabilities.
>
> **For Weakness 5: "There are some further areas of improvement ... with more complex relationships."**
>
> **Response:**
>
> Thank you for your valuable suggestions. Moving forward, we plan to incorporate a counterfactual generation mechanism into the memory system, which will generate user messages with different answers to the same question, thereby implementing a memory editing mechanism. Additionally, we plan to expand our dataset based on real-world data. We are considering leveraging actual recommendation datasets, automatically extracting relationships and entities through LLMs, and constructing corresponding Causal Generation Mechanisms to produce richer and more comprehensive user messages, questions, and answers.
>
> **We sincerely thank you for your time to review our paper, and we also thanks for your insightful comments, which, we believe, are very important to improve our paper. We hope our responses can address your concerns. If you have further questions, we are very happy to discuss them.**
>
> References:
>
> [1] Wu, Di, et al. "Longmemeval: Benchmarking chat assistants on long-term interactive memory." *arXiv:2410.10813* (2024).

---

> > ### Comment · Reviewer_GeDE · 2025-08-04
> >
> > Thanks for the detailed response, which indeed addresses my concerns.

---

> > > ### Author Response · Authors · 2025-08-04
> > >
> > > Dear reviewer GeDE,
> > >
> > > Thanks very much for your kind reply. We believe your comments are very important to improve our paper. If our responses have alleviated your concerns, is it possible to consider adjusting your score?
> > >
> > > We sincerely thank you for your time in reviewing our paper and our responses.

---

> > > > ### Comment · Reviewer_rBqV · 2025-08-08
> > > > **re**
> > > >
> > > > noted. need to provide final justification after discussions with AC and other reviewers.

---

> > > > > ### Author Response · Authors · 2025-08-09
> > > > >
> > > > > Dear Reviewer,
> > > > >
> > > > > Thank you for your response and for noting our clarifications. We completely understand that the final decision requires discussion with the AC and other reviewers, and we appreciate your thorough consideration of our work.
> > > > >
> > > > > Best regards,
> > > > >
> > > > > Submission 4675 Authors

---

### Official Review · Reviewer_CQvC · 2025-07-08

**Clarity:** 3
**Significance:** 4
**Originality:** 4
**Rating:** 4
**Confidence:** 3

**Summary:**

This paper introduces MemSim, a Bayesian simulator that automatically generates diverse and scalable question-answer pairs. These pairs are designed for evaluating the memory of LLM-based personal assistants. The authors highlight the lack of objective and automatic evaluation methods for this task.

MemSim uses a Bayesian Relation Network (BRNet) and a causal generation mechanism. This helps reduce hallucinations from LLMs when creating factual information. Based on MemSim, a new dataset called MemDaily is created, simulating daily-life scenarios. The paper provides a benchmark using MemDaily to evaluate different memory mechanisms in LLM-based agents. The authors state their work is the first to offer an objective and automatic way to evaluate LLM memory.

**Questions:**

1.  How do variations in the prompts used for data generation (e.g., user messages, facts, questions) impact the quality and characteristics of the generated dataset? Furthermore, what is the impact of using different base LLMs?
2.  Could the proposed MemSim methodology be adapted to generate memory-specific evaluation data based on existing datasets, rather than purely synthetic ones?
3.  In the Bayesian Relation Network (BRNet), is the directed graph structure manually designed (e.g., based on pre-defined rules) or is it automatically learned/generated? If the attribute set is very large and causal relationships are complex, would the graph construction process become challenging?

**Ethical Concerns:**

["NO or VERY MINOR ethics concerns only"]

**Final Justification:**

The rebuttal addressed most of my earlier concerns. My main reservation is about long-term impact: with rapidly advancing LLMs and abundant real-world data in industry, the necessity of such synthetic datasets may be limited. I therefore keep my score of 4.

**Limitations:**

yes

**Paper Formatting Concerns:**

The text font in Figure 1 is too small.

**Quality:**

3

**Strengths And Weaknesses:**

### Strengths

1. Addresses a Key Gap: This paper fills a critical void by introducing an objective and automatic evaluation method for LLM-based personal assistant memory.
2. Novel, Scalable Data Generation: MemSim, with its BRNet and causal mechanism, offers an innovative approach to automatically and scalably generate reliable evaluation data, mitigating LLM hallucinations.
3. Valuable Dataset & Open-Source: The new MemDaily dataset is a significant contribution. Its open-source release promotes reproducibility and future research.

### Weaknesses

1. Limited Cross-Dataset Comparison: The paper lacks explicit comparisons with existing public datasets, making it difficult to fully contextualize performance. While the authors note the lack of previous datasets for this specific task, comparison with less related existing datasets would also be valuable.
2. Insufficient Detail on Data Generation Prompts: The paper does not provide the specific prompts used for data generation, nor does it analyze the sensitivity of data reliability to different LLMs or prompting strategies.

---

> ### Author Rebuttal · Authors · 2025-07-29
>
> Dear reviewer CQvC,
>
> Thanks so much for your precious time in reading and reviewing our paper, and we are encouraged by your positive feedback. In the following, we try to alleviate your concerns one by one:
>
>
>
> **For Weakness 1: "Limited Cross-Dataset Compariso... related existing datasets would also be valuable."**
>
> **Response:**
>
> Thanks for your valuable question. As for the unique advantages compared with existing datasets, there are two critical advantages in our work:
>
> **(1) Automatic Data Generation without Human Annotation (Compared with Other Long-term QA Datasets):**
>
> Previous approaches usually adopt to the pipeline like "message --> question --> answer". They generate or collect some user messages, and then let an LLM generate questions based on these messages. Finally, they make the LLM generate correct answers based on the user messages and questions. Although this method is simple, the accuracy of the answers depends on the performance of the LLM, which **makes the difficulty of constructing and solving the Q&A the same**. Therefore, these approaches require further human annotation to check whether the answer is correct, such as PerLTQA, LOCOMO, and LeMon.
>
> In contrast, our proposed approach takes the pipeline like "prior knowledge --> question & answer --> message". We generate questions and answers based on constructed prior information (such as user attributes). Then, we create user messages by injecting answers with other information. This construction method makes it easier to construct Q&A than to solve them. By this means, we can ensure the correct answer is contained and well-located in the user messages.
>
> The feature of "automatic" makes the evaluation extendable to other specific scenarios without expensive human annotators.
>
> **(2) User Messages as Information Foundations for** **QAs** **(Compared with Other KBQA Datasets)**
>
> In Conventional KBQAs evaluations, a knowledge graph is typically provided as retrieval support [2], LLMs can also be evaluated on general knowledge using common-sense questions, such as HotpotQA. However, for LLM-based personal assistants, users do not provide a knowledge graph to the personal assistant. Instead, these scenarios need to convey factual information in the form of user messages. This makes it challenging to directly evaluate LLM-based agents using existing KBQA data, as it requires reliably injecting structured information into user messages. That is also the problem that our causal generation mechanism aims to address.
>
> **(3) Evaluation on memorizing certain critical information (Compared with Other Memory-based Conversation Tasks)**
>
> Some previous works like [1] focus on utilizing a long/short memory to improve long-term conversation tasks. These works can reflect "the effectiveness of memory mechanisms for long-term conversation tasks" by improving their performances on these tasks, but not take a common and direct evaluation on "**how memory mechanisms can memorize certain critical information**", which is the key point in our work. The task improvement by memory usage is not identical to the performance that the memory can exactly memorize critical information.
>
>
>
> **For Weakness 2: "Limited Cross-Dataset Compariso... related existing datasets would also be valuable."**
>
> **Response:**
>
> Thank you for your feedback. Due to our data construction mechanism's complexity and dependency on specific question types, we've included prompts in our code repository (anonymous link in abstract's last line).
>
> For MemDaily, we used rule-based mechanisms to constrain user profile generation, creating realistic BRNets reflecting real-world situations. Rather than relying solely on fixed templates, we integrate dynamic rule constraints (e.g., fathers are typically ~24 years older than children, using normal distribution). We constructed five question types for comprehensive memory evaluation: Single-hop, Multi-hop, Comparative, Aggregative, and Postprocessing. Each employs different sampling mechanisms and prompts during Causal Generation, adding prompt construction complexity.
>
> Regarding prompting strategy impacts on data generation, we conducted comparative analysis in Sections 4.1-4.2 (Tables 3-4). We tested various strategies including SeqPL (sequential entity attribute generation, similar to MDP methods) and JointPL (simultaneous generation of all attributes), applying different constraint levels to compare generation accuracy and sensitivity. Results show MemSim's prompting strategy achieves best performance with reliable data quality.
>
> For LLM impact on data generation, we tested Llama 3 and GPT-4. Llama 3 showed poor stability and failed to produce reliable data in our scenarios. While GPT-4 demonstrated excellent performance and stable, reliable data generation, its closed-source nature makes it expensive and time-consuming, leading us not to adopt it. We selected GLM-4 as our final backbone due to superior data reliability. Future work will investigate data reliability across various LLMs and scenarios.
>
>
>
> **For Question 1: "How do variations in the prompts used for data generation (e.g., user messages, facts, questions) impact the quality and characteristics of the generated dataset? Furthermore, what is the impact of using different base LLMs?"**
>
> **Response:**
>
> Thank you for this important question. We conducted comprehensive comparative analysis of different prompting strategies' impact on data generation in Sections 4.1 and 4.2.
>
> For user profile generation, we compared four strategies: (1) JointPL: generating attributes jointly; (2) SeqPL: generating attributes sequentially based on previous ones; (3) IndePL: generating attributes independently; and (4) MemSim, our BRNet-based approach. Table 3 shows MemSim outperforms other baselines on R-Human, demonstrating BRNet's effectiveness in generating high-quality user profiles.
>
> For user message generation, we evaluated four constraint strategies: (1) ZeroCons: no attribute constraints; (2) PartCons: partial attribute constraints; (3) SoftCons: full attribute constraints without enforcement; and (4) MemSim, our Causal Generation Mechanism. Figure 3 shows our method maintains high scores despite rigorous constraints on constructing reliable question-answer pairs. MemSim exhibits the highest diversity index due to BRNet and the causal generation mechanism producing varied user messages from hierarchical user profiles.
>
> Regarding different base LLMs, we tested Llama 3 and GPT-4. Llama 3 showed poor stability and failed to produce reliable quality data. GPT-4 performed excellently and generated stable, reliable data, but being closed-source, proved expensive and time-consuming. We selected GLM-4 as our backbone for superior data reliability and cost-effectiveness. Future research will investigate data reliability across various LLMs and scenarios.
>
>
>
> **For Question 2: "Could the proposed MemSim methodology be adapted to generate memory-specific evaluation data based on existing datasets, rather than purely synthetic ones?"**
>
> **Response:**
>
> Thank you for this insightful question. Yes, the proposed MemSim methodology can indeed be adapted to generate memory-specific evaluation data based on existing datasets, rather than relying solely on synthetic generation.
>
> You can leverage existing datasets by extracting entities and relationships using LLMs or other methods, then construct a BRNet following the approach described in Section 3.2 of our paper. Once the BRNet is established, you can perform sampling and generation on this network to create corresponding user messages, questions, and answers that are grounded in the original dataset.
>
> In fact, we have successfully applied MemSim to real-world business scenarios and achieved promising results, which demonstrates the practical adaptability and effectiveness of our methodology beyond purely synthetic data generation.
>
>
>
> **For Question 3: "In the Bayesian Relation Network (BRNet) ... automatically learned/generated ... challenging?"**
>
> **Response:**
>
> Thank you for this excellent question. BRNet construction supports both rule-based and automatic approaches, each suited for different scenarios. Rule-based construction works well when relationships are clear and simple with well-defined rules. It produces reliable, low-noise data since causal relationships are explicitly defined using domain knowledge. However, it becomes challenging and unscalable when attribute sets are large and causal relationships grow complex. For complex scenarios, we use LLM-based automatic extraction to identify entities and relationships for autonomous BRNet construction. This approach handles large-scale scenarios and cases where explicit rules cannot be easily designed, capturing complex relational patterns that are difficult to extract manually. The automatic approach may introduce noise during LLM extraction, requiring human verification and correction to ensure accuracy. This creates a trade-off between scalability and precision when selecting the construction method. When attribute sets are very large with complex causal relationships, automatic construction becomes the appropriate choice over rule-based methods.
>
>
>
> **For Paper Formatting Concerns: "The text font in Figure 1 is too small."**
>
> **Response:**
>
> Thank you for your valuable feedback. We will increase the font size in Figure 1 in the revised version of the paper.
>
>
>
> **We sincerely thank you for your time to review our paper, and we also thanks for your insightful comments, which, we believe, are very important to improve our paper. We hope our responses can address your concerns. If you have further questions, we are very happy to discuss them.**
>
>
>
> References:
>
> [1] Li, Hao, et al. "Hello Again! LLM-powered Personalized Agent for Long-term Dialogue." arXiv preprint arXiv:2406.05925 (2024).
>
> [2] Lan, Yunshi, et al. "Complex knowledge base question answering: A survey." IEEE TKDE 35.11 (2022): 11196-11215.

---

> > ### Comment · Reviewer_CQvC · 2025-08-05
> >
> > Thank you for your response, I will keep my score.

---

> > > ### Author Response · Authors · 2025-08-06
> > >
> > > Dear reviewer CQvC,
> > >
> > > Thanks very much for your feedback. If you have further questions, we are very happy to discuss more about them.
> > >
> > > We sincerely thank you for your time in reviewing our paper and our responses.
> > >
> > > Sincerely,
> > >
> > > Submission 4675 Authors

---

### Comment · Area_Chair_qxFC · 2025-08-05
**Friendly Reminder to Acknowledge or Update Your Review**

Dear Reviewers,

Thank you for your time and effort in reviewing the submissions and for providing valuable feedback to the authors.

If you haven’t already done so, we kindly remind you to review the authors’ rebuttals and respond accordingly. In particular, if your evaluation of the paper has changed, please update your review and explain the revision. If not, we would appreciate it if you could acknowledge the rebuttal by clicking the “Rebuttal Acknowledgement” button at your earliest convenience.

This step ensures smooth communication and helps us move forward efficiently with the review process.

We sincerely appreciate your dedication and collaboration.

Best regards, AC

---

### Note · Authors · 2025-08-12

Dear Area Chairs and Reviewers,

We would like to express our sincere gratitude for your valuable time on our submission.

In this paper, we propose MemSim, a Bayesian simulator designed to automatically construct reliable QA pairs while maintaining diversity and scalability. We introduce BRNet and a causal generation mechanism, generate the MemDaily Dataset, and conduct comprehensive validation experiments. Additionally, we provide a benchmark for evaluating agent memory mechanisms.

During the rebuttal period, we have systematically addressed each reviewer's concerns and incorporated the additional experiments they suggested. **We are pleased to report that three reviewers have positive ratings for our submission.** Reviewer rBqV mentioned in the comments that it would raise the score to 4. Reviewer GeDE also indicated it would consider increasing the rating after discussions to provide final justification. We believe their concerns are satisfactorily resolved without further questions.

However, regarding **Reviewer k9DJ**, despite our detailed responses to each of the concerns, this reviewer **did not engaged in any discussions throughout the entire process**. This reviewer remained unresponsive even after reminders from both authors and Area Chairs. **Besides, this reviewer clicked the "Rebuttal Acknowledgement" button after the discussion period had ended, but failed to meet the stated requirement of "I have engaged in discussions and responded to authors." A low score of 2 was assigned to our paper by this reviewer, and we believe there are some misunderstandings regarding our methodology and contributions** that we had hoped to clarify through the rebuttal process. For example, the limitations of evolving memory and implicit memory in Weakness 1.

We understand this may be a busy period and that reviewers might be occupied with rebuttals for their own papers, but we still hope for reviewer participation in discussions. **Therefore, we respectfully request that the Area Chairs make the final decision based on the opinions of participating reviewers, potentially assigning lower weight to or excluding this particular review. Furthermore, we strongly oppose such irresponsible reviewing behavior, which we believe should be discouraged.**

Once again, we sincerely thank the reviewers and Area Chairs for your dedication to maintaining the quality of our academic community and for your consideration of our work.

Best regards,

Submission 4675 Authors

---

### Decision · Program_Chairs · 2025-09-17

**Decision:**

Accept (poster)

**Comment:**

The paper proposes MemSim, a Bayesian simulator designed to automatically construct reliable QAs from generated user messages, with the introduction of the Bayesian Relation Network (BRNet) and a causal generation mechanism to mitigate the impact of LLM hallucinations on factual information, facilitating the automatic creation of an evaluation dataset. Therefore, the paper builds MemDaily and conducts comprehensive evaluation on this dataset to evalaute the various memory mechanisms of LLMs, including full memory, recent memory, and retrieved memory, measuring both effectiveness and efficiency.

Strengths:

1. The pipeline is automatic and easily be extended to other domains or datasets.
2. The proposed method derives user messages and QAs from the same hints sampled from BRNet. This method effectively reduces hallucination (` CQvC`, `GeDE`).
3. The authors propose a comprehensive QA taxonomy, including the 6 QA types (simple, conditional, comparative, aggregative, post-processing, and noisy). This taxonomy covers a wide range of reasoning scenarios, which provides a better and more detailed evaluation guideline for memory mechanisms (`k9DJ`, `GeDE`).

Weakness:

1. Some details are not clear, including data generation prompts (`CQvC`, `GeDE`), human evaluation (`k9DJ`, `GeDE`).
2. The synthesized data can not well reflect the real-data distritbuion (`GeDE`), some generated questions are still very artificial.
3. Lack of analysis on the impact of model size and different model family as the paper mainly conduct experiments on GLM-4-9B (` CQvC`, `GeDE`).
4. The paper does not consider complex memory applications, i.e., memory updating or more agentic components such as reflection, planning and tool use (`GeDE`, `rBqV`).

Generally, the author provide detailed response during rebuttal and all reviewers acknowledged that most of concerns are addressed.